# Causal Influence Detection for Improving Efficiency in Reinforcement Learning

**Maximilian Seitzer**
MPI for Intelligent Systems
Tübingen, Germany
`maximilian.seitzer@tue.mpg.de`

**Bernhard Schölkopf**
MPI for Intelligent Systems
Tübingen, Germany
`bs@tue.mpg.de`

**Georg Martius**
MPI for Intelligent Systems
Tübingen, Germany
`georg.martius@tue.mpg.de`

## Abstract

Many reinforcement learning (RL) environments consist of independent entities that interact sparsely. In such environments, RL agents have only limited influence over other entities in any particular situation. Our idea in this work is that learning can be efficiently guided by knowing when and what the agent can influence with its actions. To achieve this, we introduce a measure of *situation-dependent causal influence* based on conditional mutual information and show that it can reliably detect states of influence. We then propose several ways to integrate this measure into RL algorithms to improve exploration and off-policy learning. All modified algorithms show strong increases in data efficiency on robotic manipulation tasks.

## 1 Introduction

Reinforcement learning (RL) is a promising route towards versatile and dexterous artificial agents. Learning from interactions can lead to robust control strategies that can cope with all the intricacies of the real world that are hard to engineer correctly. Still, many relevant tasks such as object manipulation pose significant challenges for RL. Although impressive results have been achieved using simulation-to-real transfer [1] or heavy physical parallelization [2], training requires countless hours of interaction. Improving sample efficiency is thus a key concern in RL. In this paper, we approach this issue from a causal inference perspective.

When is an agent in control of its environment? An agent can only influence the environment by its actions. This seemingly trivial observation has the underappreciated aspect that the causal influence of actions is *situation dependent*. Consider the simple scenario of a robotic arm in front of an object on a table. Clearly, the object can only be moved when contact between the robot and object is made. Generally, there are situations where immediate causal influence is possible, while in others, none is. In this work, we formalize this situation-dependent nature of control and show how it can be exploited to improve the sample efficiency of RL agents. To this end, we derive a measure that captures the causal influence of actions on the environment and devise a practical method to compute it.

Knowing when the agent has control over an object of interest is important both from a learning and an exploration perspective. The learning algorithm should pay particular attention to these situations because (i) the robot is initially rarely in control of the object of interest, making training inefficient, (ii) physical contacts are hard to model, thus require more effort to learn and (iii) these states are enabling manipulation towards further goals. But for learning to take place, the algorithm first needs data that contains these relevant states. Thus, the agent has to take its causal influence into account already during exploration.

We propose several ways in which our measure of causal influence can be integrated into RL algorithms to address both the exploration, and the learning side. For exploration, agents can be rewarded with a bonus for visiting states of causal influence. We show that such a bonus leads the agent to quickly discover useful behavior even in the absence of task-specific rewards. Moreover,

35th Conference on Neural Information Processing Systems (NeurIPS 2021), virtual.

our approach allows to explicitly guide the exploration to favor actions with higher predicted causal impact. This works well as an alternative to $\epsilon$-greedy exploration, as we demonstrate. Finally, for learning, we propose an off-policy prioritization scheme and show that it reliably improves data efficiency. Each of our investigations is backed by empirical evaluations in robotic manipulation environments and demonstrates a clear improvement of the state-of-the-art with the same generic influence measure.

## 2 Related Work

The idea underlying our work is that an agent can only sometimes influence its surroundings. This rests on two basic assumptions about the causal structure of the world. The first is that the world consists of independent entities, in accordance with the principle of independent causal mechanisms (ICM) [3], stating that the world's generative process consists of autonomous modules. The second assumption is that the potential influence that entities have over other entities is localized spatially and occurs sparsely in time. We can see this as explaining the sparse mechanism shift hypothesis, which states that naturally occurring distribution shifts will be due to local mechanism changes [4]. This is usually traced back to the ICM principle, i.e. that interventions on one mechanism will not affect other mechanisms [5]. But we argue that it is also due to the *limited interventional range* of agents (or, more generally, physical processes), which restricts the breadth and frequency of mechanism-changes in the real world. Previous work has used sparseness to learn disentangled representations [6, 7], causal models [8], or modular architectures [9]. In the present work, we show that taking the localized and sparse nature of influence into account can also strongly improve RL algorithms.

Detecting causal influence, informally, means deciding whether changing a causal variable would have an impact on another variable. This involves causal discovery, that is, finding the existence of arrows in the causal graph [10]. While the task of causal discovery is unidentifiable in general [11], there are assumptions which permit discovery [12], in particular in the time series setting we are concerned with [13]. Even if the existence of an arrow is established, the problem remains of quantifying its causal impact, for which various measures such as transfer entropy or information flow have been proposed [14–18]. We compare how our work relates to these measures in Sec. 4.1.

The intersection of RL and causality has been the subject of recent research [19–23]. Close to ours is the work of Pitis et al. [24], who also use influence detection, albeit to create counterfactual data that augments the training of RL agents. In Sec. 5, we find that our approach to action influence detection performs better than their heuristic approach. Additionally, we demonstrate that influence detection can also be used to help agents explore better. To this end, we use influence as a type of intrinsic motivation. For exploration, various signals have been proposed, e.g. model surprise [25–27], learning progress [27, 28], empowerment [29, 30], information gain [31–33], or predictive information [34, 35]. Inspired by causality, Sontakke et al. [36] introduce an exploration signal that leads agents to experiment with the environment to discover causal factors of variation. In concurrent work, Zhao et al. [37] propose to use mutual information between the agent and the environment state for exploration. As in our work, the agent is considered a separate entity from the environment. However, their approach does not discriminate between individual situations the agent is in. Causal influence is also related to the concept of contingency awareness from psychology [38], that is, the knowledge that one's actions can affect the environment. On Atari games, exploring through the lens of contingency awareness has led to state-of-the-art results [39, 40].

## 3 Background

We are concerned with a Markov decision process $\langle \mathcal{S}, \mathcal{A}, P, r, \gamma \rangle$ consisting of state and action space, transition distribution, reward function and discount factor.[1] Most real world environments consist of entities that behave mostly independently of each other. We model this by assuming a known state space factorization $\mathcal{S} = \mathcal{S}_1 \times \ldots \times \mathcal{S}_N$, where each $\mathcal{S}_i$ corresponds to the state of an entity.

---

[1]We use capital letters (e.g. $X$) to denote random variables, small letters to denote samples drawn from particular distributions (e.g. $x \sim P_X$), and caligraphy letters to denote graphs, sets and sample spaces (e.g. $x \in \mathcal{X}$). We denote distributions with $P$ and their densitites with $p$.

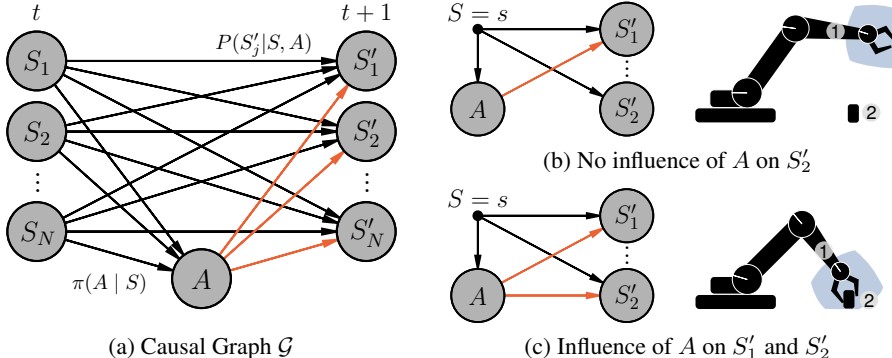

(a) Causal Graph $\mathcal{G}$      (b) No influence of $A$ on $S_2'$

(c) Influence of $A$ on $S_1'$ and $S_2'$

Figure 1: Causal graphical model capturing the environment transition from state $S$ to $S'$ by action $A$, factorized into state components. (a): Viewed globally over all time steps, all components of the state and the action can influence all state components at the next time step. (b, c): Given a situation $S = s$, some influences may or may not not hold in the *local causal graph* $\mathcal{G}_{S=s}$. In this paper, our aim is to detect which influence the action has on $S'$, i.e. the presence of the red arrows.

### 3.1 Causal Graphical Models

We can model the one-step transition dynamics at time step $t$ using a *causal graphical model* (CGM) [3, 10] over the set of random variables $\mathcal{V} = \{S_1, \ldots, S_N, A, S_1', \ldots, S_N'\}$, consisting of a directed graph $\mathcal{G}$ (see Fig. 1a) and a conditional distribution $P(V_i \mid \mathrm{Pa}(V_i))$ for each node $V_i \in \mathcal{V}$, where $\mathrm{Pa}(V_i)$ is the set of parents of $V_i$ in the causal graph. We assume that the joint distribution $P_\mathcal{V}$ is Markovian with respect to the graph [3, Def. 6.21 (iii)], that is, its density exists and factorizes as

$$p(v_1, \ldots, v_{|\mathcal{V}|}) = \prod_{i=1}^{|\mathcal{V}|} p(v_i \mid \mathrm{Pa}(V_i)). \tag{1}$$

In a CGM, we can model a (stochastic) intervention $\mathrm{do}(V_i := q(v_i \mid \mathrm{Pa}(V_i)))$ on variable $V_i$ by replacing its conditional $p(v_i \mid \mathrm{Pa}(V_i))$ in Eq. 1 with the distribution $q(v_i \mid \mathrm{Pa}(V_i))$ [3]. Here, $V_i$ could e.g. be a state component $S_i$, or the agent's action $A$. Thus, whereas a probabilistic graphical model represents a single distribution, a CGM represents a set of distributions [4].

The causal graph that we assume is shown in Fig. 1a. Within a time step, there are no edges, i.e. no instantaneous effects, except for the action which is computed by the policy $\pi(A \mid S)$. Between time steps, the graph is fully connected. The reason is that whenever an interaction between two components $S_i$ and $S_j$, however unlikely, is possible, it is necessary to include an arrow $S_i \to S_j'$ (and vice versa). Nevertheless, during most concrete time steps, there should be *no* interaction between entities, reflecting the assumption that the state components represent independent entities in the world. In particular, the agent's "sphere of influence" (depicted in blue in Figs. 1b and 1c) is limited – its action $A$ can only sparsely affect other entities. Thus, in this paper, we are interested in inferring the influence the action has in a specific state configuration $S = s$, that is, the *local causal model* in $s$.

**Definition 1.** Given a CGM with distribution $P_\mathcal{V}$ and graph $\mathcal{G}$, we define the *local CGM* induced by observing $X = x$ with $X \subset \mathcal{V}$ to be the CGM with joint distribution $P_{\mathcal{V}|X=x}$ and the graph $\mathcal{G}_{X=x}$ resulting from removing edges from $\mathcal{G}$ until $P_{\mathcal{V}|X=x}$ is causally minimal with respect to the graph.

Causal minimality tells us that each edge $X \to Y$ in the graph must be "active", in the sense that $Y$ is conditionally dependent on $X$ given all other parents of $Y$ [3, Prop. 6.36].

### 3.2 The Cause of an Effect

When is an agent's action $A = a$ the cause of an outcome $B = b$? Answering this question precisely is surprisingly non-trivial and is studied under the name of *actual causation* [10, 41]. Humans would answer by contrasting the actual outcome to some normative world in which $A = a$ did not happen, i.e. they would ask the counterfactual question "What would have happened normally to $B$ without $A = a$?" [41]. Algorithmitizing this approach poses certain problems. First, it requires a "normal" outcome which can be difficult to compute as it depends on the behavior of the different actors in

the world. Second, it requires to actually observe the world's state without the agents interference. Such a "no influence" action may not be available for every agent. Instead, we are inspired by an alternative approach, the so-called *"but-for"* test: "$B = b$ would not have happened but for $A = a$." In other words, $A = a$ was a necessary condition for $B = b$ to occur, and under a different value for $A$, $B$ would have had a different value as well. This matches well with an algorithmic view on causation: $A$ is a cause of $B$ if the value of $A$ is required to determine the value of $B$ [42].

The but-for test yields potentially counterintuitive assessments. Consider a robotic arm close to an object but performing an action that moves it away from the object. Then this action is considered a cause for the position of the object in this step, as an alternative action touching the object would have led to a different outcome. Algorithmically, knowing the action is required to determine what happens to the object – all actions are considered to be a cause in this situation. Importantly, this implies that we cannot differentiate whether individual actions are causes or not, but can only identify whether or not the agent has causal influence on other entities in the current state.

## 4   Causal Influence Detection

As the previous discussion showed, having causal influence is dependent on the situation the agent is in, rather than the chosen actions. We characterize this as the agent being *in control*, analogous to similar notions in control theory [43]. Formally, using the causal model introduced in Sec. 3, *we define the agent to be in control of $S'_j$ in state $S = s$ if there is an edge $A \to S'_j$ in the local causal graph $\mathcal{G}_{S=s}$ under all interventions $\mathrm{do}(A := \pi(a|s))$ with $\pi$ having full support.* The following proposition states when such an edge exists (proofs in Suppl. A.1).

**Proposition 1.** *Let $\mathcal{G}_{S=s}$ be the graph of the local CGM induced by $S = s$. There is an edge $A \to S'_j$ in $\mathcal{G}_{S=s}$ under the intervention $\mathrm{do}(A := \pi(a|s))$ if and only if $S'_j \not\perp\!\!\!\perp A \mid S = s$.*

To detect when the agent is in control, we can intervene with a policy. The following proposition gives conditions under which conclusions drawn from one policy generalize to many policies.

**Proposition 2.** *If there is an intervention $\mathrm{do}(A := \pi(a|s))$ under which $S'_j \not\perp\!\!\!\perp A \mid S = s$, this dependence holds under* all *interventions with full support, and the agent is in control of $S'_j$ in $s$. If there is an intervention $\mathrm{do}(A := \pi(a|s))$ with $\pi$ having full support under which $S'_j \perp\!\!\!\perp A \mid S = s$, this independence holds under* all *possible interventions and the agent is not in control of $S'_j$ in $s$.*

### 4.1   Measuring Causal Action Influence

Our goal is to find a state-dependent quantity that measures whether the agent is in control of $S'_j$. As Prop. 1 tells us, control (or its absence) is linked to the independence $S'_j \perp\!\!\!\perp A \mid S = s$. A well-known measure of dependence is the conditional mutual information (CMI) [44] which is zero for independence. We thus propose to use (pointwise) CMI as a measure of *causal action influence* (CAI) that can be thresholded to get a classification of control (see Suppl. A.2 for a derivation):

$$C^j(s) := I(S'_j; A \mid S = s) = \mathbb{E}_{a \sim \pi}\left[ \mathrm{D_{KL}}\left( P_{S'_j|s,a} \,\big\|\, P_{S'_j|s} \right) \right]. \tag{2}$$

We want this measure to be independent of the particular policy used in the joint distribution $P(S, A, S')$. This is because we might not be able to sample from or evaluate this policy (e.g. in off-policy RL, the data stems from a mixture of different policies). Fortunately, Prop. 2 shows that to detect control, it is sufficient to demonstrate (in-)dependence for a single policy with full support. Thus, we can choose a uniform distribution over the action space as the policy: $\pi(A) := \mathcal{U}(\mathcal{A})$.

Let us discuss how CAI relates to previously suggested measures of (causal) influence. *Transfer entropy* [14] is a non-linear extension of Granger causality [45] quantifying causal influence in time series under certain conditions. CAI is similar to a one-step, local transfer entropy [17] with the difference that CAI conditions on the full state $S$. Janzing et al. [18] put forward a measure of *causal strength* fulfilling several natural criteria that other measures, including transfer entropy, fail to satisfy. In Suppl. A.3, we show that CAI is a pointwise version of Janzing et al.'s causal strength, for policies not conditional on the state $S$ (adding further justification for the choice of a uniform random policy). Furthermore, we can relate CAI to notions of *controllability* [43]. Decomposing $C^j(s)$ as $H(S'_j \mid s) - H(S'_j \mid A, s)$, where $H$ denotes the conditional entropy [44], we can interpret CAI

as quantifiying the degree to which $S'_j$ can be controlled in $s$, accounting for the system's intrinsic uncertainty that cannot be reduced by the action.

In the context of RL, *empowerment* [29, 30, 46] is a well-known quantity used for intrinsically-motivated exploration that leads agents to states of maximal influence over the environment. Empowerment, for a state $s$, is defined as the channel capacity between action and a future state, which coincides with $\max_\pi C(s)$ for one-step empowerment. CAI can thus be seen as a non-trivial lower bound of empowerment that is easier to compute. However, CAI differs from empowerment in that it does not treat the state space as monolithic and is specific to an entity. In Sec. 6.1, we demonstrate that an RL agent maximizing CAI quickly achieves control over its environment.

## 4.2 Learning to Detect Control

Estimating CMI is a hard problem on many levels: it involves computing high dimensional integrals, representing complicated distributions and having access to limited data; strictly speaking, each conditioning point $s$ is seen only once in continuous spaces. In practice, one thus has to resort to an approximation. Non-parametric estimators based on nearest neighbors [47, 48] or kernels methods [49] are known to not scale well to higher dimensions [50]. Instead, we approach the problem by learning neural network models with suitable simplifying assumptions.

Expanding the KL divergence in Eq. 2, we can write CAI as

$$C^j(s) = I(S'_j; A \mid S = s) = \mathbb{E}_{A|s}\mathbb{E}_{S'_j|s,a}\left[\log \frac{p(s'_j \mid s,a)}{\int p(s'_j \mid s,a)\,\pi(a)\mathrm{d}a}\right] \tag{3}$$

To compute this term, we estimate the transition distribution $p(s'_j \mid s,a)$ from data. We then approximate the outer expectation and the transition marginal $p(s'_j \mid s)$ by sampling $K$ actions from the policy $\pi$. This gives us the estimator

$$\hat{C}^j(s) = \frac{1}{K}\sum_{i=1}^{K}\left[\mathrm{D_{KL}}\left(p(s'_j \mid s,a^{(i)}) \,\Big\|\, \frac{1}{K}\sum_{k=1}^{K}p(s'_j \mid s,a^{(k)})\right)\right], \tag{4}$$

with $\{a^{(1)},\dots,a^{(K)}\} \overset{\text{iid}}{\sim} \pi$. Here, we replaced the infinite mixture $p(s'_j \mid s)$ with a finite mixture, $p(s'_j \mid s) \approx \frac{1}{K}\sum_{i=1}^{K}p(s'_j \mid s,a^{(i)})$, and used Monte-Carlo to approximate the expectation. Poole et al. [51] show that this estimator is a lower bound converging to the true mutual information $I(S'_j; A \mid S = s)$ as $K$ increases (assuming, however, the true density $p(s'_j \mid s,a)$).

To compute the KL divergence itself, we make the simplifying assumption that the transition distribution $p(s'_j \mid s,a)$ is normally distributed given the action, which is reasonable in the robotics environment we are targeting. This allows us to estimate the KL without expensive MC sampling by using an approximation for mixtures of Gaussians from Durrieu et al. [52]. We detail the exact formula we use in Suppl. A.4.

With the normality assumption, the density itself can be learned using a probabilistic neural network and simple maximum likelihood estimation. That is, we parametrize $p(s'_j \mid s,a)$ as $\mathcal{N}(s'_j; \mu_\theta(s,a), \sigma_\theta^2(s,a))$, where $\mu_\theta, \sigma_\theta^2$ are the outputs of a neural network $f_\theta(s,a)$. We find the parameters $\theta$ by minimizing the negative log-likelihood over samples $\mathcal{D} = \{(s^{(i)}, a^{(i)}, s'^{(i)})\}_{i=1}^{N}$ collected by some policy (the univariate case shown here also extends to the multivariate case):

$$\theta^* = \arg\min_\theta \frac{1}{N}\sum_{i=1}^{N}\frac{\left(s'^{(i)}_j - \mu_\theta\big(s^{(i)}, a^{(i)}\big)\right)^2}{2\sigma_\theta^2\big(s^{(i)}, a^{(i)}\big)} + \frac{1}{2}\log\sigma_\theta^2\big(s^{(i)}, a^{(i)}\big). \tag{5}$$

There are some intricacies regarding the policy that collects the data for model training and the sampling policy $\pi$ that is used to compute CAI. First of all, the two policies need to have overlapping support to avoid evaluating the model under actions never seen during training. Furthermore, if the data policy is different from the sampling policy $\pi$, the model is biased to some degree. This suggests to use $\pi$ for collecting the data; however, as we use a random policy, this will not result in interesting data in most environments. The bias can be reduced by sampling actions from $\pi$ during data collection with some probability and only train on those. In practice, however, we find to obtain better performing models by training on all data despite potentially being biased.

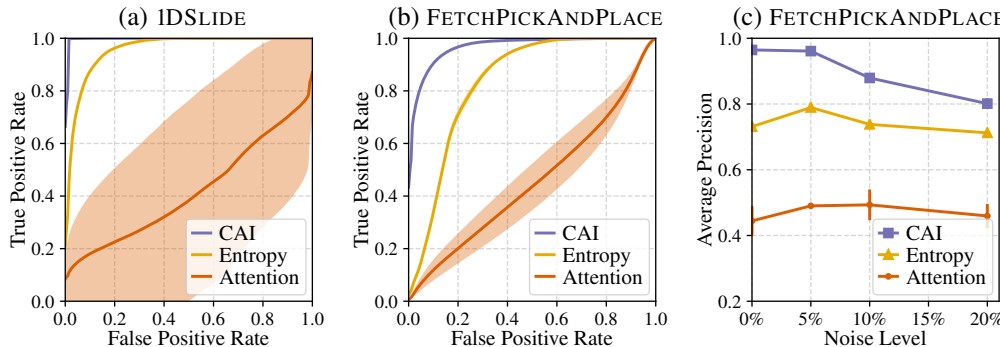

Figure 2: Causal influence detection performance. (a, b) ROC curves on 1DSLIDE and FETCH-PICKANDPLACE environments. (c) Average precision for FETCHPICKANDPLACE depending on added state noise. Noise level is given as percentage of one standard deviation over the dataset.

Table 1: Results for evaluating causal influence detection on different environments. We measure area under the ROC curve (AUC), average precision (AP), and the best achievable F-score ($F_1$).

| | 1DSLIDE | | | FETCHPICKANDPLACE | | |
|---|---|---|---|---|---|---|
| | AUC | AP | $F_1$ | AUC | AP | $F_1$ |
| CAI (ours) | $1.00 \pm 0.00$ | $0.98 \pm 0.00$ | $0.95 \pm 0.01$ | $0.97 \pm 0.01$ | $0.96 \pm 0.00$ | $0.89 \pm 0.00$ |
| Entropy | $0.96 \pm 0.00$ | $0.47 \pm 0.01$ | $0.50 \pm 0.01$ | $0.84 \pm 0.00$ | $0.73 \pm 0.00$ | $0.78 \pm 0.00$ |
| Attention [24] | $0.42 \pm 0.31$ | $0.13 \pm 0.14$ | $0.18 \pm 0.17$ | $0.46 \pm 0.06$ | $0.44 \pm 0.04$ | $0.62 \pm 0.00$ |
| Contacts | $0.89$ | $0.78$ | $0.88$ | $0.79$ | $0.77$ | $0.73$ |

## 5 Empirical Evaluation of Causal Influence Detection

In this section, we evaluate the quality of our proposed causal influence detection approach in relevant environments. As a simple test case, we designed an environment (1DSLIDE) in which the agent must slide an object to a goal location by colliding with it. Furthermore, we test on the FETCHPICKANDPLACE environment from OpenAI Gym [53], in its original setting and when adding Gaussian noise to the observations to simulate more real-world conditions. In both environments, the target variables of interest are the coordinates of the object. Note that we need the true causal graph at each time step for the evaluation. For 1DSLIDE, we derive this information from the simulation. For the pick and place environment with its non-trivial dynamics, we resort to a heuristic of the possible movement range of the robotic arm in one step. Detailed information about the setup is provided in Suppls. B and E.

For our method, we use CAI estimated according to Eq. 4 (with $K = 64$) as a classification score that is thresholded to gain a binary decision. We compare with a recently proposed method [24] that uses the attention weights of a Transformer model [54] to model influence. Moreover, we compare with an *Entropy* baseline that uses $H(S'_j \mid s)$ as a score and a *Contact* baseline based on binary contact information from the simulator. We show the test results over 5 random seeds in Table 1 and Fig. 2. We observe that CAI is able to reliably detect causal influence and no other baseline is able to do so. When increasing the observation noise, the performance drops gracefully for CAI as shown in Fig. 2c. Suppl. C contains more experimental results, including a visualization of CAI's behavior.

## 6 Improving Efficiency in Reinforcement Learning

Having established the efficacy of our causal action influence (CAI) measure, we now develop several approaches to use it to improve RL algorithms. We will empirically verify the following claims in robotic manipulation environments: CAI improves sample efficiency and performance by (i) better state exploration through an exploration bonus, (ii) causal action exploration, and (iii) prioritizing experiences with causal influence during training.

We consider the environments FETCHPUSH, FETCHPICKANDPLACE from OpenAI Gym [55], and FETCHROTTABLE which is our modification containing a rotating table (explained in Suppl. B.3).

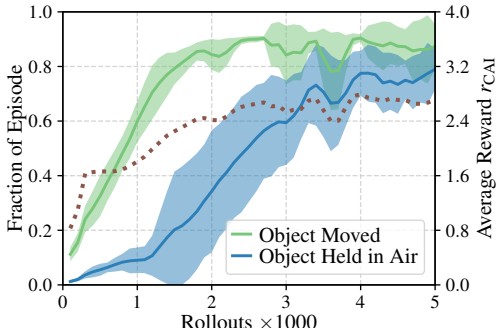

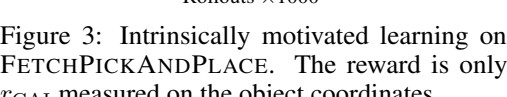

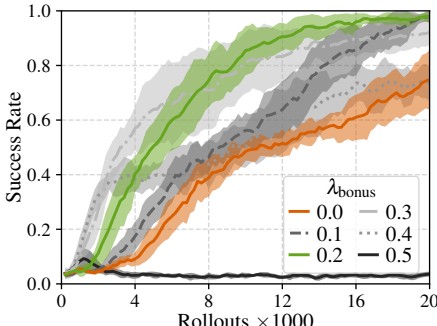

Figure 3: Intrinsically motivated learning on FETCHPICKANDPLACE. The reward is only $r_{\text{CAI}}$ measured on the object coordinates.

Figure 4: *Exploration bonus* improves performance in FETCHPICKANDPLACE. Sensitivity to the bonus reward scale $\lambda_{\text{bonus}}$.

These environments are goal-conditioned RL tasks with sparse rewards, meaning that each episode, a new goal is provided and the agent only receives a distinct reward upon reaching it. We use DDPG [56] with hindsight experience replay (HER) [57] as the base RL algorithm, a combination that achieves state-of-the-art results in these environment. The influence detection model is trained online on the data collected from an RL agent learning to solve its task. Since our measure $C^j$ requires an entity of interest, we choose the coordinates of the object (as $S_j$). In all experiments, we report the mean success rate with standard deviation over 10 random seeds. More information about the experimental settings can be found in Suppl. F.

## 6.1 Intrinsic Motivation to Seek Influence

**Causal Action Influence as Reward Bonus.** We hypothesize that it is useful for an agent to be intrinsically motivated to gain control over its environment. We test this hypothesis by letting the agent maximize the causal influence it has over entities of interest. This can be achieved by using our influence measure as a reward signal. The reward signal can be used on its own, as an intrinsic motivation-type objective, or in conjunction with a task-specific reward as an exploration bonus. In the former case, we expect the agent to discover useful behaviors that can help it master task-oriented skills afterwards; in the latter case, we expect learning efficiency to improve, especially in sparse extrinsic reward scenarios. Concretely, for a state $s$, we define the bonus as $r_{\text{CAI}}(s) = C^j(s)$, and the total reward as $r(s) = r_{\text{task}}(s) + \lambda_{\text{bonus}} r_{\text{CAI}}(s)$, where $r_{\text{task}}(s)$ is the task reward, and $\lambda_{\text{bonus}}$ is a hyperparameter.

**Experiment on Intrinsically Motivated Learning.** We first test the behavior of the agent in the absence of any task-specific reward on the FETCHPICKANDPLACE environment. Interestingly, the agent learned to grasp, lift, and hold the object in the air already after 2000 episodes, as shown in Fig. 3. The results demonstrate that encouraging causal control over the environment is well suited to prepare the agent for further tasks it might have to solve.

**Impact of CAI Reward Bonus.** Second, we are interested in the impact of adding an exploration bonus. In Fig. 4, we present results on the FETCHPICKANDPLACE environment when varying the reward scale $\lambda_{\text{bonus}}$. Naturally, the exploration bonus needs to be selected in the appropriate scale as a value too high will make it dominate the task reward. If selected correctly, the sample efficiency is improved drastically; for example, we find that the agent reaches a success rate of 60% four-times faster than the baseline (DDPG+HER) without any bonus ($\lambda_{\text{bonus}} = 0$).

## 6.2 Actively Exploring Actions with Causal Influence

**Following Actions with the Most Causal Influence.** Exploration via bonus rewards favors the re-visitation of already seen states. An alternative approach to exploration uses pro-active planning to choose exploratory actions. In our case, we can make use of our learned influence estimator to pick actions which we expect will have the largest causal effect on the agent's surroundings. From a causal viewpoint, the resulting agent can be seen as an experimenter that performs planned interventions in the environment to verify its beliefs. Should the actual outcome differ from the expected outcome, subsequent model updates can integrate the new data to self-correct the causal influence estimator.

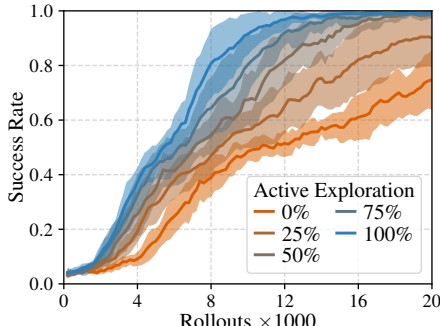
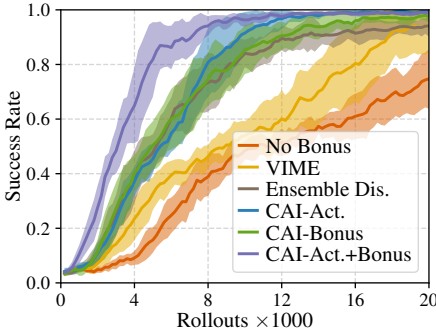

Figure 5: Performance of *active exploration* in FETCHPICKANDPLACE depending on the fraction of exploratory actions chosen actively (Eq. 6) from a total of 30% exploratory actions.

Figure 6: Experiment comparing exploration strategies on FETCHPICKANDPLACE. The combination of active exploration and reward bonus yields the largest sample efficiency.

Concretely, given the agent being in state $s$, we choose the action that has the largest contribution to the empirical mean in Eq. 4:

$$a^* = \underset{a \in \{a^{(1)}, \dots, a^{(K)}\}}{\arg\max} \mathrm{D_{KL}}\left( p(s'_j \mid s, a) \,\Big\|\, \frac{1}{K} \sum_{k=1}^{K} p(s'_j \mid s, a^{(k)}) \right), \tag{6}$$

with $\{a^{(1)}, \dots, a^{(K)}\} \overset{\text{iid}}{\sim} \pi$. Intuitively, the selected action will be the one which results in maximal deviation from the expected outcome under all actions. For states $s$ where the the agent is not in control, i.e. $C^j(s) \approx 0$, the action selection is uniform at random.

**Active Exploration in Practice.** Can active exploration replace $\epsilon$-greedy exploration? To gain insights, we study the impact of the fraction of actively chosen exploration actions. For every exploratory action ($\epsilon$ is 30% in our experiments), we choose an action according to Eq. 6 the specified fraction of the time, and otherwise a random action. Figure 5 shows that any amount of active exploration improves over simple random exploration. Active causal action exploration can improve the sample efficiency roughly by a factor of two.

**Combined CAI Exploration.** We also present the combination of reward bonus and active exploration and compare our method with VIME, another exploration scheme based on information-theoretic measures [33]. In contrast to our method, VIME maximizes the information gain about the state transition dynamics. Further, we compare to ensemble disagreement [58], which in effect minimizes epistemic uncertainty about the transition dynamics. We compare different variants of VIME and ensemble disagreement in Suppl. D, and display only their best versions here. Figure 6 quantifies the superiority of all CAI variants (with ensemble disagreement as a viable alternative) and shows that combining the two exploration strategies compounds to increase sample efficiency even further. In the figure, CAI uses 100% active exploration and $\lambda_{\text{bonus}} = 0.2$ as the bonus reward scale.

### 6.3  Causal Influence-based Experience Replay

**Prioritizing According to Causal Influence.** We will now propose another method using CAI, namely to inform the choice of samples replayed to the agent during off-policy training. Typically, past states are sampled uniformly for learning. Intuitively, focusing on those states where the agent has control over the object of interest (as measured by CAI) should improve the sample efficiency. We can implement this idea using a prioritization scheme that samples past episodes in which the agent had more influence more frequently. Concretely, we define the probability $P^{(i)}$ of sampling any state from episode $i$ (of $M$ episodes) in the replay buffer as

$$P^{(i)} = \frac{p^{(i)}}{\sum_{i=1}^{M} p^{(i)}} \cdot \frac{1}{T}, \qquad \text{with} \qquad p^{(i)} = \left( M + 1 - \underset{i}{\mathrm{rank}} \sum_{t=1}^{T} C^j\big(s^{(t)}\big) \right)^{-1}. \tag{7, 8}$$

where $T$ is the episode length, and $p^{(i)}$ is the priority of episode $i$. The priority of an episode $i$ is based on the (inverse) rank of the episode ($\mathrm{rank_i}$) when sorting all $M$ episodes according to their total influence (i.e. sum of state influences). We call this *causal action influence prioritization* (CAI-P).

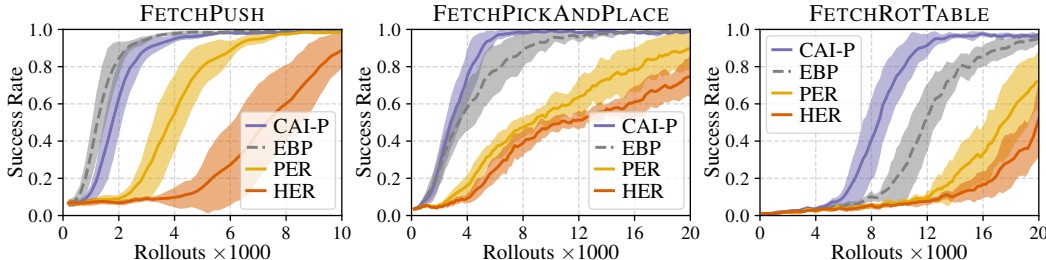

Figure 7: Prioritizing experience replay in different manipulation environments. Comparison of causal action influence prioritization (CAI-P) against baselines: the energy-based method (EBP) [60] with privileged information, prioritized experience replay (PER) [59], and HER without prioritization.

This scheme is similar to Prioritized Experience Replay [59], with two differences: instead of using the TD error for prioritization, we use the causal influence measure. Furthermore, instead of prioritizing individual states, we prioritize episodes and sample states uniformly within episodes. This is because the information about the return that can be achieved from an influence state still needs to be propagated back to non-influenced states by TD updates, which requires sampling them.

**Influence-Based Prioritization in Manipulation Tasks.** We compare our influence-based prioritization (CAI-P) against no prioritization in hindsight experience replay (HER) (a strong baseline for multi-goal RL), and two other prioritization schemes: prioritized experience replay (PER) [59] and energy-based prioritization (EBP) [60]. Especially EBP is a strong method for the environments we are considering as it uses privileged knowledge of the underlying physics to replay episodes based on the amount of energy that is transferred from agent to the object to manipulate. All prioritization variants are equipped with HER as well. The FETCHROTTABLE environment, shown in Fig. 8, is an interesting test bed as the object can move through the table rotation without the control of the agent. The results, shown in Fig. 7, reveal that causal influence prioritization can speed up learning drastically. Our method is on par or better than the energy-based (oracle) method EBP and improves over PER by a factor of 1.5–2.5 in learning speed (at 60% success rate). Finally, in Suppl. D, we combine all our proposed improvements and show that FETCHPICKANDPLACE can be solved up to 95% success rate in just 3000 episodes.

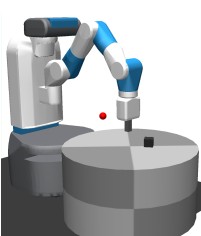

Figure 8: FETCH ROTTABLE. The table rotates periodically.

## 7 Discussion

In this work, we show how situation-dependent causal influence detection can help improve reinforcement learning agents. To this end, we derive a measure of local causal action influence (CAI) and introduce a data-driven approach based on neural network models to estimate it. We showcase using CAI as an exploration bonus, as a way to perform active action exploration, and to prioritize in experience replay. Each of our applications yields strong improvements in sample efficiency. We expect that there are further ways to use our causal measure in RL, e.g. for credit assignment.

Our work has several limitations. First, we assume full observability of the state, which simplifies the causal inference problem as there is no confounding between an agent's action and its effect. Under partial observability, our approach could still be applicable using latent variable models [61]. Second, we require an available factorization of the state into causal variables. The problem of automatically learning causal variables from high-dimensional data is open [4] and our method would likely benefit from advances in this field. Third, the accurate estimation of our measure relies on a correct model. We found that deep networks can struggle at times to pick up the causal relationship between actions and entities. How to design models with appropriate inductive biases for cause-effect inference is an open question [3, 4, 62].

An intriguing future direction is to extend our work to influence detection between entities, a prerequisite for identifying multi-step influences of the agent on the environment. Being able to model such indirect interventions would bring us closer to "artificial scientists" – agents that can perform planned experiments to reveal the latent causal structure of the world.

## Acknowledgments and Disclosure of Funding

The authors thank Andrii Zadaianchuk and Dominik Zietlow for many helpful discussions and providing feedback on the text. Furthermore, the authors would also like to thank Sebastian Blaes for creating the FETCHROTTABLE environment. The authors thank the International Max Planck Research School for Intelligent Systems (IMPRS-IS) for supporting Maximilian Seitzer. GM and BS are members of the Machine Learning Cluster of Excellence, EXC number 2064/1 – Project number 390727645. We acknowledge the support from the German Federal Ministry of Education and Research (BMBF) through the Tübingen AI Center (FKZ: 01IS18039B).

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
