# Supplementary Material

## A  Proofs and Derivations

### A.1  Proof of Propositions

We first clarify the behavior of local CGMs (see Def. 1) under interventions. In particular, the local CGM induced by $X = x$ on $P(\mathcal{V})$ under an intervention $\mathrm{do}(V := v)$ is defined to be the local CGM induced by $X = x$ on the joint distribution $P^{\mathrm{do}(V:=v)}(\mathcal{V})$.

*Proof of Prop. 1.* "if": If it holds that $S'_j \not\perp\!\!\!\perp A \mid S = s$, then by the Markov property [3, Def. 6.21], there must be an unblocked path from $A$ to $S'_j$ in $\mathcal{G}_{S=s}$. Because the path over $S$ is blocked by observing $S = s$, and we assume no instantaneous effects, the only possible such path is the direct edge $A \to S'_j$.
"only if": If there is an edge $A \to S'_j$ in $\mathcal{G}_{S=s}$, then causal minimality of the local causal graph says that $S'_j$ must be dependent on each parent given its other parents, meaning $S'_j \not\perp\!\!\!\perp A \mid S = s$.  □

*Proof of Prop. 2.* To show the first part, note that the dependence $S'_j \not\perp\!\!\!\perp A \mid S = s$ under $\mathrm{do}(A := \pi(a|s))$ implies that there exists some $s'_j$ and $a_1$, $a_2$ with $\pi(a_1 \mid s) > 0$, $\pi(a_2 \mid s) > 0$ for which

$$p^{\mathrm{do}(A:=\pi)}(s'_j \mid s, a_1) = p(s'_j \mid s, a_1) \neq p(s'_j \mid s, a_2) = p^{\mathrm{do}(A:=\pi)}(s'_j \mid s, a_2). \tag{9}$$

Any $\pi'$ with full support would also have $\pi'(a_1 \mid s) > 0$, $\pi'(a_2 \mid s) > 0$, and so

$$p^{\mathrm{do}(A:=\pi')}(s'_j \mid s, a_1) = p(s'_j \mid s, a_1) \neq p(s'_j \mid s, a_2) = p^{\mathrm{do}(A:=\pi')}(s'_j \mid s, a_2), \tag{10}$$

implying the dependence under $\mathrm{do}(A := \pi')$. To show that the agent is in control of $S'_j$ in $S = s$, there needs to be an edge $A \to S'_j$ in $\mathcal{G}_{S=s}$ under all interventions $\mathrm{do}(A := \pi')$ with $\pi'$ having full support. This is the case, because as shown, for all interventions $\mathrm{do}(A := \pi')$ with $\pi'$ having full support it holds that $S'_j \not\perp\!\!\!\perp A \mid S = s$, and by Prop. 1, there is an edge $A \to S'_j$ in $\mathcal{G}_{S=s}$ under any such intervention.

To show the second part, we show that if $S'_j \perp\!\!\!\perp A \mid S = s$ under any intervention $\mathrm{do}(A := \pi)$ with $\pi$ having full support, for any intervention $\mathrm{do}(A := \pi')$ it holds that $P^{\mathrm{do}(A:=\pi')}(S'_j \mid S = s, A) = P^{\mathrm{do}(A:=\pi')}(S'_j \mid S = s)$. This follows from the fact that for any $\pi'$, it holds that

$$P^{\mathrm{do}(A:=\pi')}(S'_j \mid S = s, A) = P(S'_j \mid S = s, A) \tag{11}$$

$$= P(S'_j \mid S = s) = P^{\mathrm{do}(A:=\pi')}(S'_j \mid S = s) \tag{12}$$

where the first equality is due to the autonomy property of causal mechanisms [3, Eq. 6.7], and the second equality because of the independence $S'_j \perp\!\!\!\perp A \mid S = s$. Note that if $\pi$ had not had full support, we would not be allowed to use the second equality as then $P(S'_j \mid S = s, A = a) = P(S'_j \mid S = s)$ only for $a$ with $\pi(a) > 0$. The agent is not in control of $S'_j$ in $S = s$, as for all interventions $\mathrm{do}(A := \pi')$ with $\pi'$ having full support, there is no edge $A \to S'_j$ in $\mathcal{G}_{S=s}$ by Prop. 1.  □

### A.2  CMI Formula

$$C^j(s) = I(S'_j; A \mid S = s) = D_{\mathrm{KL}}\Big( P_{S'_j, A|s} \,\big\|\, P_{S'_j|s} \otimes P_{A|s} \Big) \tag{13}$$

$$= \mathbb{E}_{S'_j, A|s}\left[ \log \frac{p(s'_j, a \mid s)}{p(s'_j \mid s)\pi(a \mid s)} \right] \tag{14}$$

$$= \mathbb{E}_{S'_j, A|s}\left[ \log \frac{p(s'_j \mid s, a)}{p(s'_j \mid s)} \right] \tag{15}$$

$$= \mathbb{E}_{A|s}\Big[ D_{\mathrm{KL}}\Big( P_{S'_j|s,a} \,\big\|\, P_{S'_j|s} \Big) \Big] \tag{16}$$

## A.3 Proof that CAI is a Pointwise Version of Janzing et al.'s Causal Strength

Janzing et al.'s [18] measure of causal strength quantifies the impact that removing a set of arrows in the causal graph would have. As it is the relevant case for us, we concentrate here on the single arrow version, for instance, between random variables $X$ and $Y$. The idea is to consider the arrow as a "communication channel" and evaluate the corruption that could be done to the signal that flows between $X$ and $Y$ by cutting the channel. To do so, the distribution that feeds the channel is replaced with $P(X)$, i.e. the marginal distribution of $X$. The measure of causal strength then is equal to the difference between the pre- and post-cutting joint distribution as given by the KL divergence.

Formally, let $\mathcal{V}$ denote the set of variables in the causal graph, let $X \to Y$ be the arrow of interest with $X, Y \in \mathcal{V}$, and let $\mathrm{Pa}_Y^{\backslash X}$ be the set of parents of $Y$ without $X$. Then, the post-cutting distribution on $Y$ is defined as

$$p_{X \to Y}\left(y \mid \mathrm{pa}_Y^{\backslash X}\right) = \int p\left(y \mid x, \mathrm{pa}_Y^{\backslash X}\right) p(x) \mathrm{d}x. \tag{17}$$

The new joint distribution after such an intervention is defined as

$$p_{X \to Y}\left(v_1, \ldots, v_{|\mathcal{V}|}\right) = p_{X \to Y}\left(y \mid \mathrm{pa}_Y^{\backslash X}\right) \prod_{\substack{j \\ v_j \neq y}} p(v_j \mid \mathrm{Pa}(v_j)). \tag{18}$$

Finally, the causal strength of the arrow $X \to Y$ is defined as

$$\mathfrak{C}_{X \to Y} = \mathrm{D}_{\mathrm{KL}}(P(V) \| P_{X \to Y}(V)). \tag{19}$$

To show the correspondence to our measure, we first restate Lemma 3 of [18]. For a single arrow $X \to Y$, causal strength can also be written as the KL between the conditionals on $Y$:

$$\mathfrak{C}_{X \to Y} = \mathrm{D}_{\mathrm{KL}}\left( P(Y \mid \mathrm{Pa}_Y) \| P_{X \to Y}\left(Y \mid \mathrm{Pa}_Y^{\backslash X}\right)\right) \tag{20}$$

$$= \int \mathrm{D}_{\mathrm{KL}}\left( P(Y \mid \mathrm{pa}_Y) \| P_{X \to Y}\left(Y \mid \mathrm{pa}_Y^{\backslash X}\right)\right) P(\mathrm{pa}_Y) \mathrm{dpa}_Y \tag{21}$$

We rewrite this further:

$$= \int P\left(\mathrm{pa}_Y^{\backslash X}\right) P\left(X \mid \mathrm{pa}_Y^{\backslash X}\right) \mathrm{D}_{\mathrm{KL}}\left( P(Y \mid \mathrm{pa}_Y) \| P_{X \to Y}\left(Y \mid \mathrm{pa}_Y^{\backslash X}\right)\right) \mathrm{d}x \, \mathrm{dpa}_Y^{\backslash X} \tag{22}$$

$$= \mathbb{E}_{\mathrm{pa}_Y^{\backslash X}}\left[ \int P\left(X \mid \mathrm{pa}_Y^{\backslash X}\right) \mathrm{D}_{\mathrm{KL}}\left( P(Y \mid \mathrm{pa}_Y) \| P_{X \to Y}\left(Y \mid \mathrm{pa}_Y^{\backslash X}\right)\right) \mathrm{d}x \right] \tag{23}$$

$$= \mathbb{E}_{\mathrm{pa}_Y^{\backslash X}}\left[ \mathbb{E}_{x \mid \mathrm{pa}_Y^{\backslash X}}\left[ \mathrm{D}_{\mathrm{KL}}\left( P(Y \mid \mathrm{pa}_Y) \| P_{X \to Y}\left(Y \mid \mathrm{pa}_Y^{\backslash X}\right)\right)\right]\right] \tag{24}$$

And we see that the inner part corresponds to our measure $C^j(s)$ (cmp. Eq. 2) for the choices of variables $X \triangleq A$, $Y \triangleq S_j'$, and $\mathrm{Pa}_Y^{\backslash X} \triangleq S$, provided that $P(X) \triangleq \pi(A)$ is not dependent on $S$. Thus, CAI is a pointwise version of causal strength for policies independent of the state:

$$\mathfrak{C}_{A \to S_j'} = \mathbb{E}_s\left[C^j(s)\right] \qquad \square$$

## A.4 Approximating the KL Divergence Between a Gaussian and a Mixture of Gaussians

In this section, we give the approximation we use for the KL divergence in Eq. 4. We first state the approximation for the general case of the KL between two mixtures of Gaussians, and then specialize to our case when the first distribution is Gaussian distributed. Here, we use the notation of Durrieu et al. [52].

Let $f$ be the PDFs of a multivariate Gaussians mixture with $A$ components, mixture weights $\omega_a^f \in (0, 1]$, means $\mu_a^f \in \mathbb{R}^d$ and covariances $\Sigma_a^f \in \mathbb{R}^{d \times d}$, where $a \in \{1, \ldots, A\}$ is the index is of the $a$'th component. Then,

$$f(x) = \sum_{a=1}^{A} \omega_a^f f_a(x) = \sum_{a=1}^{A} \omega_a^f \mathcal{N}\left(x; \mu_a^f, \Sigma_a^f\right), \tag{25}$$

where $f_a(x) = \mathcal{N}\big(x; \mu_a^f, \Sigma_a^f\big)$ is the PDF of a multivariate Gaussian with mean $\mu_a^f$ and covariance $\Sigma_a^f$. Analously, let $g$ be the PDF of a multivariate Gaussians mixture with $B$ components. We are interested in the KL divergence $D_{\text{KL}}(f \parallel g) = \int_{\mathbb{R}^d} f(x) \log \frac{f(x)}{g(x)} dx$, which is intractable.

There are several ways to approximate the KL based on the decomposition $D_{\text{KL}} = H(f, g) - H(f)$, where $H(f, g)$ is the cross-entropy between $f$ and $g$ and $H(f)$ is the entropy of $f$. We will state the so-called *product* approximation $D_{\text{prod}}$ and the *variational* approximation $D_{\text{var}}$ [63]. Starting with $D_{\text{prod}}$:

$$H(f, g) \geq -\sum_a \omega_a^f \log\left(\sum_b \omega_b^g t_{ab}\right) \tag{26}$$

$$H(f) \geq -\sum_a \omega_a^f \log\left(\sum_{a'} \omega_{a'}^f z_{aa'}\right) \tag{27}$$

$$D_{\text{prod}} := \sum_a \omega_a^f \log\left(\frac{\sum_{a'} \omega_{a'}^f z_{aa'}}{\sum_b \omega_b^g t_{ab}}\right), \tag{28}$$

where $t_{ab} = \int f_a(x) g_b(x) dx$, $z_{aa'} = \int f_a(x) f_{a'}(x) dx$ are normalization constants of product of Gaussians, and the inequalities in (26), (27) are based on Jensen's inequality. For the variational approximation:

$$H(f, g) \leq \sum_a \omega_a^f H(f_a) - \sum_a \omega_a^f \log\left(\sum_b \omega_b^g e^{-D_{\text{KL}}(f_a \parallel g_b)}\right) \tag{29}$$

$$H(f) \leq \sum_a \omega_a^f H(f_a) - \sum_a \omega_a^f \log\left(\sum_{a'} \omega_{a'}^f e^{-D_{\text{KL}}(f_a \parallel f_{a'})}\right) \tag{30}$$

$$D_{\text{var}} := \sum_a \omega_a^f \log\left(\frac{\sum_{a'} \omega_{a'}^f e^{-D_{\text{KL}}(f_a \parallel f_{a'})}}{\sum_b \omega_b^g e^{-D_{\text{KL}}(f_a \parallel g_b)}}\right), \tag{31}$$

where (29), (30) are based on solving variational problems. It can be shown that the mean between $D_{\text{prod}}$ and $D_{\text{var}}$ is the mean of a lower and upper bound to $D_{\text{KL}}$, with better approximation qualities [52]. Consequently, we use $D_{\text{mean}} := \frac{D_{\text{prod}} + D_{\text{var}}}{2}$ as the basis of our approximation. We can simplify $D_{\text{mean}}$ as in our case, we know that $f$ has only one component. This means that we can compute $H(f)$ in closed form, and do not need to use the inequalitities (27), (30). Approximating $H(f, g)$ with the mean of the lower bound (26) and upper bound (29),

$$H_{\text{mean}}(f, g) := \frac{1}{2}\left(-\log\left(\sum_b \omega_b^g t_{ab}\right) + H(f) - \log\left(\sum_b \omega_b^g e^{-D_{\text{KL}}(f_a \parallel g_b)}\right)\right), \tag{32}$$

we get the final formula we use

$$D_{\text{mean}} := H_{\text{mean}}(f, g) - H(f) \tag{33}$$

$$= -\frac{1}{2} \log\left(\sum_b \omega_b^g t_{ab}\right) - \frac{1}{2} \log\left(\sum_b \omega_b^g e^{-D_{\text{KL}}(f_a \parallel g_b)}\right) - \frac{1}{2} H(f). \tag{34}$$

Note that this term can become negative, whereas the KL is non-negative. In practice, we thus threshold $D_{\text{mean}}$ at zero. For completeness, we also state the entropy of a Gaussian

$$H(f) = \frac{1}{2} \log\big((2\pi e)^d |\Sigma|\big), \tag{35}$$

the log normalization constant for a product of Gaussians

$$\log t_{ab} = -\frac{d}{2} \log 2\pi - \frac{1}{2} \log|\Sigma_a^f + \Sigma_b^g| - \frac{1}{2}(\mu_b^g - \mu_a^f)^T (\Sigma_a^f + \Sigma_b^g)^{-1}(\mu_b^g - \mu_a^f), \tag{36}$$

and the KL between two Gaussians

$$D_{\text{KL}}(f_a \parallel g_b) = -\frac{d}{2} + \frac{1}{2} \log \frac{|\Sigma_a^f|}{|\Sigma_b^g|} + \frac{1}{2} \text{Tr}\big((\Sigma_b^g)^{-1} \Sigma_a^f\big) + \frac{1}{2}(\mu_b^g - \mu_a^f)^T (\Sigma_b^g)^{-1}(\mu_b^g - \mu_a^f). \tag{37}$$

In our experiments, we assume independent dimensions, that is, we parametrize the covariance $\Sigma$ as a diagonal matrix. With this, the above formulas can be further simplified.

# B  Environments

## B.1  1D-Slide

1DSLIDE is a simple environment that we designed to test influence detection, as we can easily derive when the agent has influence or not. It consists of an agent and an object positioned on a line, with both agent and object only being able to move left and right. See Fig. S1 for a visualization. The goal of the agent is to move the object to a goal zone. As the agent can not cross past the center of the environment, it has to hit the object at the appropriate speed. The state space $\mathcal{S} \subset \mathbb{R}^4$ consists of position and velocity of the agent and object. The agent's action $a \in \mathcal{A} \subseteq [-1, 1]$ applies acceleration to the agent. On contact with the object, the full impulse of the agent is transferred to the object. We can derive if the agent has causal influence in a state by simulating applying the maximum acceleration in both directions and checking whether a contact has occurred, and the object state has changed.

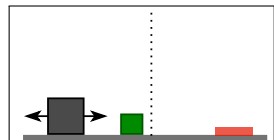

Figure S1: Schematic of 1DSLIDE environment. The agent (black square) has to slide the object (green square) to the target location (red zone), but can not pass the dotted line.

## B.2  FetchPickAndPlace

The FETCH environments in OpenAI Gym [53] are built on top of the MuJoCo simulator [64], and consist of a position-controlled 7 DoF robotic arm [55]. In FETCHPICKANDPLACE, the state space $\mathcal{S} \subset \mathbb{R}^{25}$ consists of position and velocity of the robot's end effector, the position and velocity of the gripper, and the object's position and rotation, linear and angular velocities, and position relative to the end effector. The action space $\mathcal{A} \subseteq [-1, 1]^4$ controls the gripper movement and opening/closening of the gripper. For the experiments involving Transformer models, we split the state space into agent and object state components, where we do not include the relative positions between gripper and object into either component.

For our experiment in Sec. 5 evaluating the causal influence detection, we need to determine whether the agent can potentially influence the object, i.e. whether the agent is "in control". This is difficult to determine analytically, which is why we designed a heuristic. The idea is to find an ellipsoid around the end effector that captures its maximal movement range, and intersect this ellipsoid with the object to decide whether influence is possible. As the range of movement of the robotic arm is different depending on its position, we first build a lookup table (offline) that contains, for different start positions, end positions after applying different actions. To do so, we grid-sample the 3D-space over the table (50 grid points per dimension), plus a sparser sampling of the outer regions (20 grid points per dimension), resulting in 133 000 starting locations for the lookup table. Then, the following procedure is repeated for each starting location and action: after resetting the simulator, the end effector is manually moved to one of the starting locations, one of the maximal actions in each dimension (i.e., $-1$ and $1$, for a total of 6 actions) is applied, and the end position after one environment step is recorded in the lookup table.

Now, while the environment runs, for each step, we find the sampling point closest to the position of the robotic arm in the lookup table using a KD-tree, and find the corresponding end positions. From the end positions, we build the ellipsoid by taking the maximum absolute deviation in each dimension to be the length of the ellipsoid axis in this dimension. The ellipsoid so far does not take into account the spatial extents of object and gripper. Thus, we extend the ellipsoid by the sizes of the object and gripper fingers in each dimension. Furthermore, we take into account that the gripper fingers can be slid in y-direction by the agent's action by extending the ellipsoid's y-axis correspondingly. The label of "agent in control" is then obtained by checking whether the object location lies within the ellipsoid. Last, we also label a state as "agent in control" when there is an actual contact between gripper and object in the following step. We note that the exact procedure described above is included in the code release.

## B.3  FetchRotTable

FETCHROTTABLE is an environment designed by us to test CAI prioritization in a more challenging setting. In FETCHROTTABLE, the table rotates periodically, moving the object around. This creates a confounding effect for influence detection, as there is another source of object movements besides

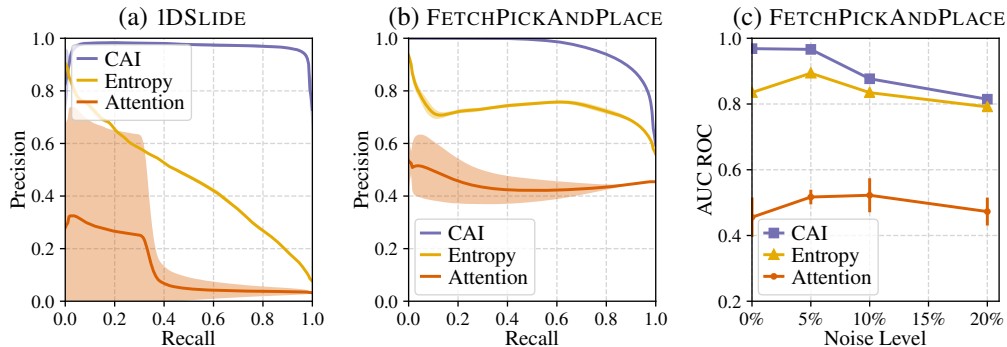

Figure S2: Causal influence detection performance, complementing Fig. 2 in the main part. (a, b) Precision-recall curves on 1DSLIDE and FETCHPICKANDPLACE environments. (c) Area-under-ROC curve for FETCHPICKANDPLACE depending on added state noise. Noise level is given as percentage of one standard deviation over the dataset.

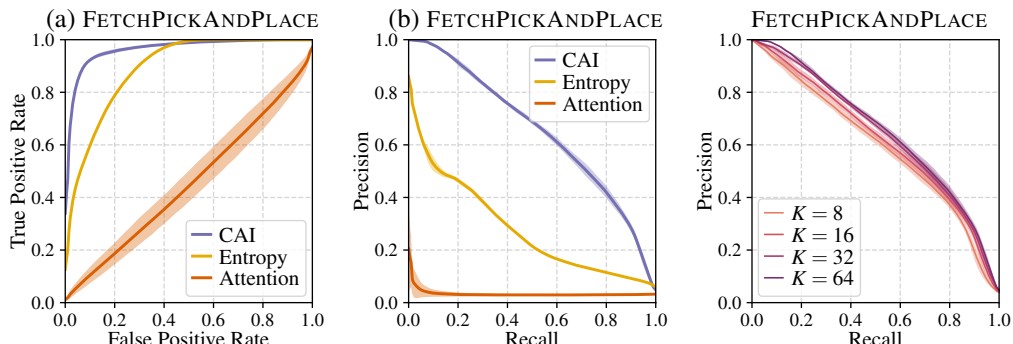

Figure S3: Causal influence detection performance on a test dataset of 250k transitions from a *random policy*. This dataset only has 3.3% transitions with influence (i.e. labeled as positive), which is why the detection task is considerably harder. Left, center: ROC and PR curves. Right: PR curves for CAI when varying the number of actions $K$.

the agent's own actions. This means that CAI needs to differentiate between different causes for movements.

FETCHROTTABLE is based on FETCHPICKANDPLACE, but the rectangular table is replaced with a circular one (see Fig. 8). The table rotates with inertia by 45 degrees over the course of 25 environment steps, and then pauses for 5 steps. To make the resulting environment Markovian, the state space of FETCHPICKANDPLACE is extended to $\mathcal{S} \subset \mathbb{R}^{29}$, additionally including sine and cosine of the table angle, the rotational velocity of the table, and the amount the table will rotate in the current state in radians. The task of the agent is the same as in FETCHPICKANDPLACE, i.e. to move the object to a target position. If the target position is on the table, the agent thus has to learn to hold the object in position while the table rotates. In contrast to FETCHPICKANDPLACE, the goal distribution is different, with 90% of goals in the air and only 10% on the table. This makes the task more challenging, as the agent has to master grasping and lifting before 90% of the possible positive rewards can be accessed.

## C  Additional Results for Influence Evaluation

In this section, we include additional results for the empirical evaluation of influence detection in Sec. 5. Figure S2a and Fig. S3b show precision-recall (PR) curves for the experiment in Sec. 5, while Fig. S2c shows how area-under-ROC curve varies while increasing the observation noise level.

For FETCHPICKANDPLACE, we also evaluated on a test dataset obtained from a random policy. On this dataset, the detection task is considerably harder: it contains only 3.3% transitions where the

agent has influence (i.e. labeled as positive), and it does not contain samples where the agent moves the object in the air, which are easier to detect as influence. We show ROC and PR curves for this dataset in Fig. S3a and Fig. S3b. As expected, the detection performance drops compared to the other test dataset, which can in particular be seen in the PR curve. But overall, the performance is still quite good when taking into account the low amount of positive samples. In Fig. S3c, we also plot the impact of varying the number of sampled actions $K$ on this dataset. As one can see, the method is relatively robust to this parameter, giving decent performance even under a small number of sampled actions. However, we think that a higher number of sampled actions is important in edge-case situations, which are overshadowed in such a quantitative analysis.

Finally, in Fig. S4, we give a qualitative analysis of CAI. Here, we plot the trajectory in three different situations: no contact of agent and object, briefly touching the object, and successfully manipulating the object. As desired, CAI is low and high in the "no contact" and "successful manipulation" trajectories. In the "brief touch" trajectory, CAI spikes around the contact. However, in steps 8-11, when the agent hovers near the object, the heuristical labeling (see Sec. B.2) still detects that influence of agent on object is possible, which CAI does not register. These are difficult cases which show that our method still has room for improvement; we think that these failure cases could be resolved by employing a better model. However, we also note that the "ground truth" label is only based on a heuristic, which will make mispredictions at times as well.

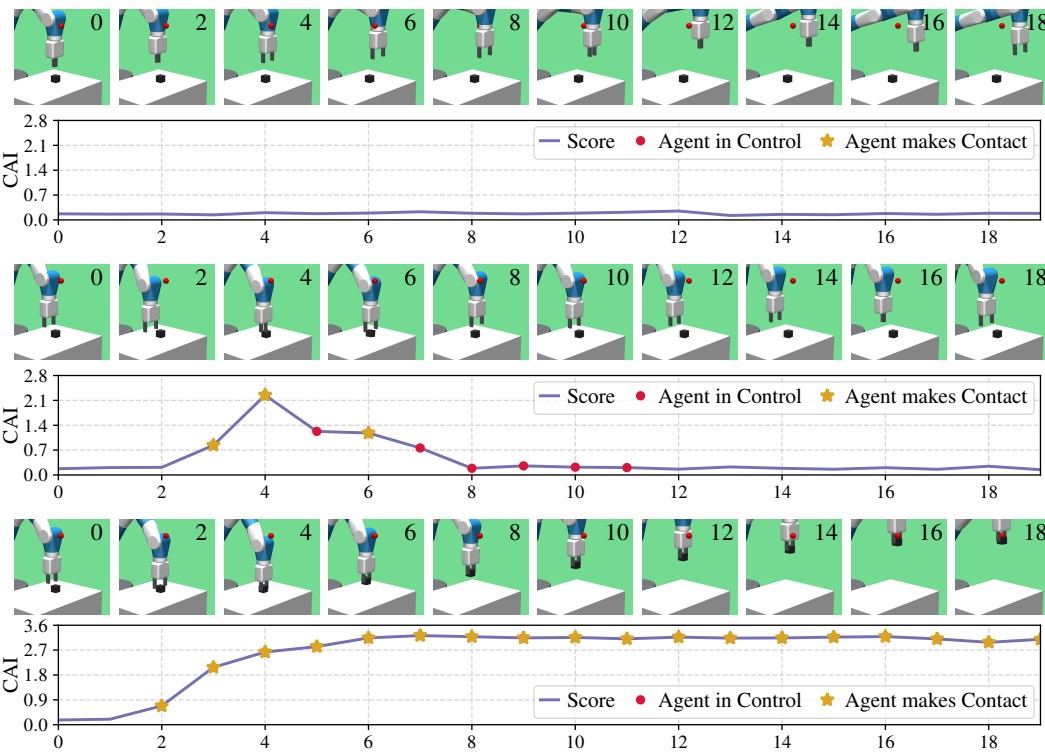

Figure S4: Visualizing CAI score $C^j(s)$ over first 20 steps of an episode on FETCHPICKANDPLACE. The red dots mark states where the agent has causal influence on the object (according to the ground truth label as described in Sec. B.2). Yellow stars mark that the agent makes contact with the object between this state and the next state (as derived from the simulator), which also includes that the agent has causal influence. Plotted are episodes with no contact (first row), briefly touching the object (second row), and successful manipulation of the object (third row).

# D    Additional Results for Reinforcement Learning

In this section, we include additional results to the RL experiments in Sec. 6. First of, in Fig. S5, we analyse CAI's behavior using one the runs from the intrinsic motivation experiment in Sec. 6.1. To this end, we plot a heatmap visualising the score distribution in the replay buffer after 5 000

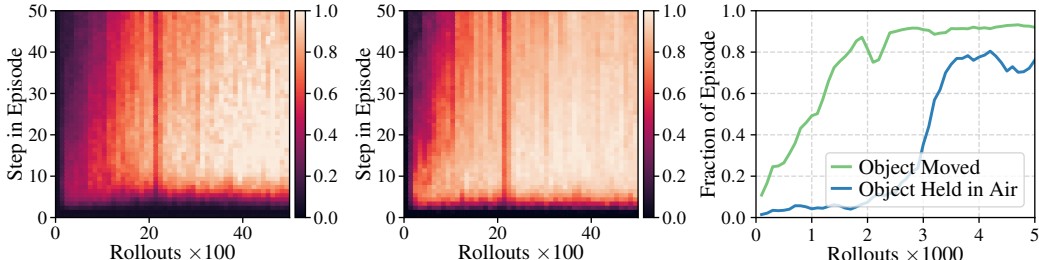

Figure S5: Analyzing the behavior of CAI. Plotted is one run of the intrinsic motivation experiment from Sec. 6.1. Left: heatmap showing score $C^j(s)$ per step, as stored in replay buffer after 5 000 episodes. Score is averaged in groups of 100 episodes and normalized by 95th percentile. Center: heatmap showing ground truth label derived for causal influence, as described in Sec. B.2. CAI approximates the ground truth well. Right: behavior of the agent in this run.

episodes, a corresponding heatmap with ground truth labels, and the fraction of steps where the object was moved or held in the air by the agent. It can be seen that CAI's distribution approximates the ground truth label's distribution well. It also becomes visible that CAI measures the *strength of causal influence*, as the score is highest after the agent learns to lift the object in the air (episode 3 000 onwards). In comparison, the binary ground truth distribution assigns high scores more uniformly.

In Fig. S6, we analyse the impact of combining our different proposed improvements, namely prioritization, exploration bonus, and active action selection. On its own, prioritization brings the largest benefit. Combining either exploration bonus or active action selection with prioritization leads to similar further improvements. Both are complementary however; combining all three variants together results in the best performing version. Compared to not using CAI, combining two or all three improvements leads to a 4–10× increase in sample efficiency.

In Fig. S5, we plot different versions of the VIME baseline. For VIME, there is the choice of which parts of the state space the information gain should be computed on. We compared the variants of using the full state, only the position of the agent and the object, and only the position of the object (which would be similar to CAI). The variant of using agent and object position performed best, and thus we use it for comparison in Fig. 6 in the main part.

Finally, in Fig. S8, we plot different versions of the ensemble disagreement baseline. Similarly to VIME, we also have the option of choosing parts of the state space the disagreement should be computed on. We compared the variants of using the full state, only the position of the agent and the object, and only the position of the object (which would be similar to CAI). The variant of using just object position performed best, and thus we use it for comparison in Fig. 6 in the main part.

**Preliminary Experiments with Negative Results**   In preliminary work, we also experimented with other variants, which we list here for completeness. For prioritization, we tested using a proportional distribution over episode scores (as in [59]) instead of the ranked distribution. While the proportional distribution also worked, it resulted in slower learning and performance did not converge to 100% success rate. We also briefly tried shaping the ranking distribution by raising the score by a power before ranking (as in [59]), but this did not result in notable improvements, so we dropped this line for simplicity. Further, Schaul et al. [59] use importance sampling to correct for the bias in state sampling introduced by the prioritization. We found this not to be necessary for the tasks we experimented with, but note that it could be required to converge for other environments. Last, we also experimented with prioritizing states within an episode, and prioritized selection of goals for HER, but could not achieve any improvements from this.

For the exploration bonus, we tested adding a flat bonus when the score is over a certain threshold, and a bonus that interacts multiplicative with the reward. Both variants performed worse than the additive bonus. Linearly annealing the bonus to zero over the course of training did also not result in improvements. For active action selection, we experimented with sampling actions according to a ranked or proportional distribution over the CAI score instead of taking the action with a maximum score. Both versions performed worse than maximum score action selection.

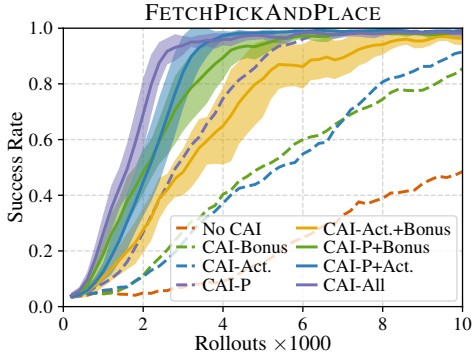

Figure S6: Performance when combining our proposed improvements, namely prioritization (P), exploration bonus (Bonus), active action selection (Act.), in FETCHPICKANDPLACE. All proposed improvements act complementary and yield compounded increase in sample efficiency. For clarity, we omit the standard deviation of the single variant versions here, which can be found in the main part.

Figure S7: Comparing different variants of the VIME baseline [33] for exploration in FETCH-PICKANDPLACE. The versions differ in the state components used for information gain. VIME-Object: object coordinates. VIME-Object+Agent: object and robotic gripper coordinates. VIME-Full: full state, including velocities and rotation state.

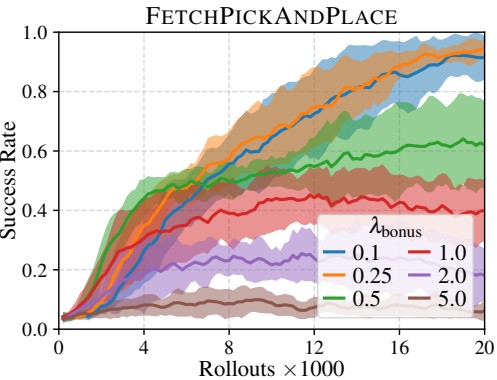
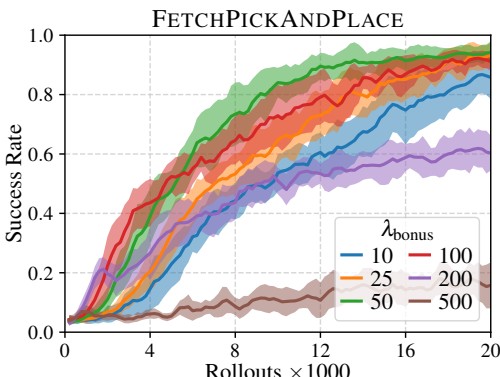

Figure S8: Comparing different variants of the Ensemble Disagreement baseline [58]. The versions differ in the state components used for computing the disagreement, and the weight of the exploration bonus $\lambda_{\mathrm{bonus}}$. Left: use full state, including velocities and rotation state, take bonus from time of collecting rollout. Right: use only object coordinates, recompute bonus after model updates.

# E   Settings of Influence Detection Evaluation

Code is available under https://github.com/martius-lab/cid-in-rl.

**Datasets**   For the experiment in Sec. 5, we use separate training, validation and test datasets. For 1DSLIDE, training and validation set consist of 1 000 episodes (30 000 samples) collected from two separate training runs of a DDPG+HER agent. The test set consists of 4 000 episodes (120 000 samples), where 2 000 episodes were collected by a random policy, and 2 000 by a DDPG+HER agent. For FETCHPICKANDPLACE, training and validation set consist of 5 000 episodes (250 000 samples) collected from two separate training runs of a DDPG+HER agent. The test set consists of 7 500 episodes (375 000 samples) collected from training a DDPG+HER agent for 30 000 epochs. The test set was subsampled by selecting 250 random episodes from every 1 000 collected episodes (subsampling was performed to ease the computational load of evaluation). The test set has 45.3% samples that are labeled positive, i.e. the agent has influence. We note that it is not strictly necessary to train on data collected from a training RL agent, as we also trained on data from random policies and obtained similar results. In Sec. 5, we also analyse the behavior when adding observation noise, which is given as a percentage level. To determine an appropriate level of noise, we recorded the

standard deviations over the different state dimensions on a dataset with 10 000 episodes collected from training a DDPG+HER agent. We then add Gaussian noise to each dimension of the state with standard deviation equal to a percentage of the dataset standard deviation.

In Fig. S3, we also give results for FETCHPICKANDPLACE on a different test set. This test set consists of 5 000 episodes (250 000 samples) collected from a random policy. It only has 3.3% samples that are labeled positive, i.e. there is considerably less interaction with the object than on the other test set.

**Methods** The classification scores for the different methods were obtained as follows. For CAI, we use $\hat{C}^j(s)$ as given in Eq. 4, with $K = 64$ actions for sampling. For entropy, we use the same model as trained for CAI, and estimate conditional entropy as $H(S'_j \mid S = s) \approx \frac{1}{K} \sum_{i=1}^{K} H\big[p(s'_j \mid s, a^{(i)})\big]$ with $\{a^{(1)}, \ldots, a^{(K)}\} \overset{\text{iid}}{\sim} \pi$, using the formula given in Eq. 35 for the entropy of a Gaussian. For attention, we use the attention weights of a Transformer, where the score is computed as follows. As the Transformer requires a set as input, we split the input vector into components for the agent state, the object state, and the action. The Transformer is trained to predict the position of the object, that is, we discard the Transformer's output for the other components. Then, letting $A_i$ denote the attention matrix of the $i$'th of $N$ layers, the total attention matrix is computed as $\prod_{i=1}^{N} A_i$. The score is the entry of the total attention matrix corresponding to the input position of the action component, and the output position of the object component. We refer to Pitis et al. [24] for more details.

We list the model hyperparameters for CAI in Table S1 and for the Transformer in Table S2. Training was performed for a maximum of 3 000 epochs on 1DSLIDE and 2 000 epochs on FETCHPICKAND-PLACE. The training was stopped early when the mean-squared-error (MSE) did not improve for 10 evaluations on the validation set, where evaluations were conducted every 20 training epochs. We trained all models using the Adam optimizer [65], with $\beta_1 = 0.9$, $\beta_2 = 0.999$ if not noted otherwise. All models were trained to predict the relative difference to the next state instead of the absolute next state, i.e. the target was $S'_j - S_j$. For FETCHPICKANDPLACE, we rescaled the targets by a factor of 50, as it resulted in better performance.

We used a simple multi-layer perceptron (MLP) for the model in CAI, with two separate output layers for mean and variance. To constrain the variance to positive range, the variance output of the MLP was processed by a softplus function (given by $\log(1 + \exp(x))$), and a small positive constant of $10^{-8}$ was added to prevent instabilities near zero. We also clip the variance to a maximum value of 200. For weight initialization, orthogonal initialization [66] was used. We observed that training with the Gaussian likelihood loss (Eq. 5) can be unstable. Applying spectral normalization [67] to some of the layers decreased the instability considerably. Note that we did not apply spectral normalization to the mean output layer, as doing so resulted in worse predictive performance. For FETCHPICKANDPLACE, the inputs were normalized by applying batch normalization (with no learnable parameters) before the first layer.

Table S1: Settings for CAI on Influence Evaluation.

(a) 1DSLIDE settings.

| Parameter | Value |
|---|---|
| Batch Size | 1000 |
| Learning Rate | 0.0003 |
| Network Size | $4 \times 128$ |
| Activation Function | ReLU |
| Spectral Norm on $\sigma$ | Yes |
| Spectral Norm on Layers | Yes |
| Normalize Input | No |

(b) FETCHPICKANDPLACE settings.

| Parameter | Value |
|---|---|
| Batch Size | 500 |
| Learning Rate | 0.0008 |
| Network Size | $3 \times 256$ |
| Activation Function | ReLU |
| Spectral Norm on $\sigma$ | Yes |
| Spectral Norm on Layers | Yes |
| Normalize Input | Yes |

# F   Settings of Reinforcement Learning Experiments

Our RL experiments are run in the goal-conditioned setting, that is, each episode, a goal is created from the environment that the agent has to reach. An episode counts as success when the goal

Table S2: Settings for Transformer on Influence Evaluation.

(a) 1DSLIDE settings.

| Parameter | Value |
|---|---|
| Batch Size | 1000 |
| Learning Rate | 0.0003 |
| Embedding Dimension | 16 |
| FC Dimension | 32 |
| Number Attention Heads | 1 |
| Number Transformer Layers | 2 |
| Normalize Input | No |

(b) FETCHPICKANDPLACE settings.

| Parameter | Value |
|---|---|
| Batch Size | 500 |
| Learning Rate | 0.0001 |
| Embedding Dimension | 128 |
| FC Dimension | 128 |
| Number Attention Heads | 2 |
| Number Transformer Layers | 3 |
| Normalize Input | Yes |

is reached upon the last step of the episode. On the FETCH environments we use, the goal are coordinates where the object has to be moved to. For goal sampling, we use the settings as given by the environments in OpenAI Gym (described in more detail by Plappert et al. [55]). We use the sparse reward setting, that is, the agent receives a reward of $0$ when the goal is reached, and $-1$ otherwise. Practically, goal-conditioned RL is implemented by adding the current goal to the input for policy and value function. For evaluating the RL experiments, we run the current RL agent 100-times every 200 episodes, and average the outcomes to obtain the success rate.

For the RL experiments, we use the same base settings for all algorithms, listed in Table S3. These settings (for DDPG+HER) were borrowed from Ren et al. [68], as they already provide excellent performance compared to the original settings from Plappert et al. [55] (e.g. on FETCHPICKAND-PLACE, $4\times$ faster to reach 90% success rate). Before starting to train the agent, the memory buffer is seeded with 200 episodes collected from a random policy. The input for policy and value function is normalized by tracking mean and standard deviation of inputs seen over the course of training.

All experiments can be run on a standard multi-core CPU machine. Training a CAI model with prioritization for 20 000 epochs on FETCHPICKANDPLACE takes roughly 8 hours using 3 cores on an Intel Xeon Gold 6154 CPU with 3 GHz. Training a DDPG+HER agent takes roughly 4 hours.

**CAI Implementation**   For CAI, we list the settings in Table S4. After an episode is collected by the agent, the CAI score $C^j$ is computed for the episode and stored in the replay buffer. The model for CAI was trained every 100 episodes by sampling batches from the replay buffer. For online training, we designed a training schedule where the number of training batches used is varied over the course of training. We note that the specific schedule used appeared to be not that important as long as the model is sufficiently trained initially. We mostly chose our training schedule to make training computationally cheaper. After every round of model training, the CAI score $C^j$ is recomputed on the states stored in the replay buffer. With larger buffer sizes, this can become time consuming. We observed that it is also possible to fully recompute scores only every 1 000 epochs, and otherwise recompute scores only on the newest 1 000 epochs, at only a small loss of performance.

**Exploration Bonus**   For the exploration bonus experiments, we clip the scores at specific values (reported in the settings tables under "Maximum Bonus") to restrict the influence of outliers before multiplying with $\lambda_{\text{bonus}}$ and addition to the task reward. Moreover, the total reward is clipped to a maximum value of 0 to prevent making states more attractive than reached-goal states.

**Baselines**   We list specific hyperparameters for the baselines VIME [33], ensemble disagreement [58], and PER [59] in Table S5. For VIME [33], we adapted a Pytorch port[2] of the official implementation[3]. For PER, we used the hyperparameters as proposed in Schaul et al. [59] and implemented it ourselves. For EBP [60], we adapted the official implementation[4].

Ensemble disagreement [58] was implemented by ourselves. We experimented with two variants. The first variant (*Full State + Direct Bonus* in Table S5) is close to the original, that is, we use the full state prediction for the bonus computation, train the model every 10 collected rollouts and

---

[2]Alec Tschantz. https://github.com/alec-tschantz/vime. MIT license.

[3]OpenAI. https://github.com/openai/vime. MIT license.

[4]Rui Zhao. https://github.com/ruizhaogit/EnergyBasedPrioritization. MIT license.

compute the bonus for each state using the current model when collecting that state. The second variant (*Object Position + Recompute Bonus* in Table S5) is closer to CAI, that is, we only use the position of the object for bonus computation, train the model every 100 collected rollouts, and fully recompute the bonus for each collected state every 1 000 epochs (see CAI implementation above). For both variants, we use the same neural network architectures as CAI and train the models using the mean squared error loss.

Table S3: Base settings for DDPG+HER. Also used for CAI, VIME, EBP, and PER.

(a) General settings.

| Parameter | Value |
|---|---|
| Episode Length | 50 |
| Batch Size | 256 |
| Updates per Episode | 20 |
| Replay Buffer Warmup | 200 |
| Replay Buffer Size | 500 000 |
| Learning Rate | 0.001 |
| Discount Factor $\gamma$ | 0.98 |
| Polyak Averaging | 0.95 |
| Action $L_2$ Penalty | 1 |
| Action Noise | 0.2 |
| Random $\epsilon$-Exploration | 0.3 |
| Observation Clipping | $[-5, 5]$ |
| Q-Target Clipping | $[-50, 0]$ |
| Policy Network | $3 \times 256$ |
| Q-Function Network | $3 \times 256$ |
| Activation Function | ReLU |
| Weight Initialization | Xavier Uniform [69] |
| Normalize Input | Yes |
| HER Replay Strategy | Future |
| HER Replay Probability | 0.8 |

(b) Environment/task-specific settings.

| FETCHROTTABLE | |
|---|---|
| Parameter | Value |
| Learning Rate | 0.0003 |
| Discount Factor $\gamma$ | 0.95 |
| Polyak Averaging | 0.99 |

| FETCHPICKANDPLACE *Intrinsic Motivation* | |
|---|---|
| Parameter | Value |
| Replay Buffer Warmup | 100 |
| Learning Rate | 0.003 |
| Discount Factor $\gamma$ | 0.95 |
| Polyak Averaging | 0.99 |
| Q-Function Network | $2 \times 192$ |
| Q-Target Clipping | None |
| HER Replay Strategy | No HER |

Table S4: Settings for CAI in RL experiments.

(a) General settings.

| Parameter | Value |
|---|---|
| Batch Size | 500 |
| Train Model Every | 100 Episodes |
| Training Schedule | |
| Initial (200 Episodes) | 40 000 Batches |
| $\leq 5\,200$ Episodes | 10 000 Batches |
| $\leq 10\,200$ Episodes | 5 000 Batches |
| $> 10\,200$ Episodes | No Training |
| Adam $\beta_2$ | 0.9 |
| Learning Rate | 0.0008 |
| Network Size | $4 \times 256$ |
| Activation Function | Tanh |
| Spectral Norm on $\sigma$ | Yes |
| Normalize Input | Yes |
| CAI Number of Actions $K$ | 32 |

(b) Environment/task-specific settings.

| FETCHROTTABLE | |
|---|---|
| Parameter | Value |
| Network Size | $4 \times 386$ |

| FETCHPICKANDPLACE *Intrinsic Motivation* | |
|---|---|
| Parameter | Value |
| Training Schedule | |
| $\leq 5\,200$ Episodes | 20 000 Batches |
| Maximum Bonus | 10 |

| *Exploration Bonus* | |
|---|---|
| Parameter | Value |
| Maximum Bonus | 2 |

Table S5: Settings for Baselines in RL experiments.

| *VIME* [33] | |
|---|---|
| **Parameter** | **Value** |
| Batch Size | 25 |
| Train Model Every | 1 Episode |
| Batches per Update | 100 |
| Learning Rate | 0.003 |
| Bayesian NN Size | $2 \times 64$ |
| Number of ELBO Samples | 10 |
| KL Term Weight | 0.05 |
| Likelihood $\sigma$ | 0.5 |
| Prior $\sigma$ | 0.1 |
| Normalize Input | Yes |
| Batch Size Info Gain | 10 |
| Update Steps Info Gain | 10 |
| Bonus Weight | 0.3 |
| Maximum Bonus | 10 |

| *Ensemble Disagreement* [58] | |
|---|---|
| **Parameter** | **Value** |
| Number of Ensembles | 5 |
| Batch Size | 500 |
| Adam $\beta_2$ | 0.9 |
| Learning Rate | 0.0008 |
| Network Size | $4 \times 256$ |
| Activation Function | Tanh |
| Normalize Input | Yes |
| *Full State + Direct Bonus* | |
| Train Model Every | 10 Episodes |
| Train For | 500 Batches |
| Maximum Bonus | 5 |
| *Object Position + Recompute Bonus* | |
| Train Model Every | 100 Episodes |
| Train For | 5 000 Batches |
| Maximum Bonus | 0.1 |

| *Prioritized Experience Replay* [59] | |
|---|---|
| **Parameter** | **Value** |
| $\alpha$ | 0.6 |
| $\beta$ | Linearly Scheduled from 0.4 to 1.0 over 20 000 Epochs |