# OpenReview forum: "Causal Influence Detection for Improving Efficiency in Reinforcement Learning"
_NeurIPS.cc/2021/Conference — NeurIPS 2021 Poster_

### Official Review · Reviewer_HRby · 2021-07-08

**Rating:** 7
**Confidence:** 4

**Summary:**

This work is built upon the idea of independent causal mechanisms in reinforcement learning environments where an agent only has limited inference over other entities in a given situation. Based on this premise, the authors introduce a measure called situation-dependent causal influence (CAI), which is conditional mutual information. The authors show how to estimate CAI and how to use it in exploration and RL policy training.


**Limitations And Societal Impact:**

The discussion is satisfactory.

**Main Review:**

I like the idea behind the work: because an RL agent has only a limited invervetional range, exploration and experience replay in RL could be more focused to be effective. At the same time, there are a couple of issues that need to be addressed/clarified.

First, it seems that simple heuristics could achieve the same goal. Specifically, the work seems to be targeting rotis environments (line 192). In this case, my feeling is that CAI identification be rather straightforward; similar to the heuristics used in the simulation (line 216) - the agent can only affect the states of the possible movement range of the robotic arm in one step. Clearly, exploration should be more in these areas. Would that suffice? When would we need the more sophisticated algorithm proposed in Sec. 4?

Second, in the current evaluation, the authors evaluate how effective their algorithms can estimate CAI in Sec. 5. Clearly, their algorithm outperforms others significantly. There are two questions. First, how about a simple heuristic, like the one mentioned above? Second, just estimating CAI itself is not the ultimate goal. The ultimate goal is training an RL agent effectively. In Figure 5, no other exploration method is compared except fixed \epsilon-greedy. In Figure 6, only VIME and its variants are compared. In the literature of effective exploration, many algorithms have been proposed.   A more thorough evaluation is needed to really demonstrate the benefit of the proposed algorithm. As shown in Figure 7, CAI has similar performance with EBP in two cases and outperforms EBP in one.

There are a few strong assumptions needed in the paper. On one hand, I understand the necessity of starting with something simple. On the other hand, I wonder how much it would be possible to relax some of these assumptions. First, the model requires a known space factorization. How critical is this assumption? In robotics environments, could this be done relatively easily using some heuristics? Second, the paper is based on the causality framework, which I greatly appreciate. However, causality has its strong assumptions. I was wondering if the work really depends on the causality framework. It seems to me that the key component is to evaluate the conditional mutual information, presented in Eq. 2. Could we start from that? That would make the work much more general.


**Time Spent Reviewing:**

3.5

---

> ### Author Response · Authors · 2021-08-09
> **Author response**
>
> We thank you for your review. You raised several valid concerns which we address in the following.
>
> First, about using a heuristic based on the agent‘s movement range. We would like to point out that deriving such a heuristic requires privileged information about how the environment works. The heuristic we used to evaluate the detection performance of CAI (line 216), uses the ground-truth simulator to approximate the possible movement range. In contrast, CAI does not require such a model of the environment. Our method is also applicable in more complicated scenarios where a simple distance-based heuristic would fail; e.g. imagine a scenario where the robotic arm is able to control another object via a joystick. Admittedly such a scenario is not very common, however, heuristics have to be manually designed per environment, whereas CAI is universally applicable. CAI can also deal with stochasticity and noisy observations, which would have to be integrated into the heuristic first.
>
> Nevertheless, as per your suggestion, we ran a variant where we used the heuristic as the exploration signal. We summarize the results in the following, and we will include a corresponding plot in the camera-ready version. As expected, using the “ground-truth” heuristic also works well to improve exploration, but is slightly worse than using CAI. We see two potential reasons for this. First, the heuristic (as it‘s based on fitted ellipsoids) may be flawed in areas where CAI can more accurately predict the behavior of the robot. Second, the heuristic is a qualitative measure („in control or not“), whereas CAI is quantitative and could thus assign more importance to some states („strongly in control“).
>
> Second, about the evaluation. We agree that another baseline for exploration would be useful and we added one based on model-based surprise. We refer to the general answer where we discuss this in more detail. Also, as discussed above, we compare against exploration using the heuristic. For prioritization with EBP, we would like to point out that this is a method that uses privileged information about the environment. As such, we consider CAI reaching similar performance as EBP, a positive result for our method.
>
> Third, about the assumption of a factorized state space. We think that in robotics environments, it is natural to start policy learning with a factorization of the world into entities that are relevant for the robot to solve its tasks. For example, in object manipulation, we typically have a goal space available that contains a factorized representation of the objects. Relaxing these assumptions (e.g. by starting from images) is an interesting direction for future work. For this, we expect that recent methods for object-centric representation learning (e.g. slot attention [1] or SCALOR [2]) could be well combined with our method. These methods structure the learned state space into objects/entities, which results in the required factorization.
>
> Last, about the causal framework. You are correct that the key quantity is the conditional mutual information (CMI). We are following a general strategy to perform causal inference: we try to reduce causal quantities (generally not estimable from observational data) to statistical quantities (which we can estimate from data). Once we have the statistical estimator, we do not „need“ the causal framework anymore in some sense. But the causal framework is what allowed us to derive the CMI in the first place, and allows us to be sure that it corresponds to the strength of the causal effect we are interested in (assuming a causal model). Another benefit of the causal framework is that it ensures that evaluating the CMI for one policy gives us valid results under alternative, counterfactual policies (cf. Proposition 2). Of course, this all relies on the assumptions of a certain causal model/graph. But we think that the causal model we are assuming is a fairly natural one for many RL environments (e.g. full observability is implied by using an MDP), and thus we believe that the method does not need very strong assumptions.
>
> [1]: F. Locatello et al., “Object-Centric Learning with Slot Attention,” Advances in Neural Information Processing Systems, vol. 33, 2020.
>
> [2]: J. Jiang, S. Janghorbani, G. de Melo, and S. Ahn, “SCALOR: Generative World Models with Scalable Object Representations,” International Conference on Learning Representations, 2020.

---

> > ### Comment · Reviewer_HRby · 2021-08-15
> > **Causual model assumption questions**
> >
> > Thanks for including the experiment and intuition on heuristics, especially CAI is more quantitative. I would like to clarify one thing. You mentioned that heuristics relies on the knowledge of how environments work while "CAI does not require such a model of the environment". I am not sure what you mean here. Does not causual model require that knowledge; i.e., how states change based on previous states and actions (Fig. 1)?
> >
> > Second, full observability, being able to observe all states, does not directly imply causal graph. For example, causal graph typically assumes minimality (Assumption 3.2 in [1]), which does not seem to be directly obtainable from full observability.  Another example is when two state variables are completely correlated, say S1=f(S2) where f is reversable, could Prop. 2 solve this issue?
> >
> > [1] Introduction to Causal Inference (ICI), from a Machine Learning Perspective. Brady Neal.

---

> > > ### Author Response · Authors · 2021-08-18
> > > **Answers to "Causual model assumption"**
> > >
> > > Thank you for asking these clarification questions. Sorry for the late reply (vacations).
> > >
> > > > You mentioned that heuristics relies on the knowledge of how environments work while "CAI does not require such a model of the environment". I am not sure what you mean here. Does not causal model require that knowledge; i.e., how states change based on previous states and actions (Fig. 1)?
> > >
> > > The heuristics would need to know how far the arm can move in one timestep and what it means to control/influence the variable of interest (the object), i.e. by touching it. This is meant by "knowledge of how the env. works".
> > > In contrast, to compute CAI we learn a generic model using a neural network. This would work for any kind of causal influence, be it based on the arm being close to an object or controlling it with a remote control, or some other mechanism.
> > > We are only supplying the generic (full graph) in Fig.1(a) to the algorithm. The concrete models in Fig 1b,c can be obtained by thresholding CAI for instance.
> > >
> > >  > Second, full observability, ....
> > >
> > > Our comment on full observability should not have implied that this gives us the causal graph. Instead, we intended to bring across that with full observability we do not need to estimate latent factors.
> > > The assumptions are that we have a general graph as in Fig 1a, where we know what are our actions and the corresponding temporal order of all elements, but this is very natural in RL.
> > >
> > > > Another example is when two state variables are completely correlated, say S1=f(S2) where f is reversable, could Prop. 2 solve this issue?
> > >
> > > If $S_1$ and $S_2$ refer to two components of the state-vector and are at the same time-step then, if $A$ has causal influence on $S_1$ then also on $S_2$ and vice versa, but then the relationship should be $S_1=f(S_2,A)$.
> > > If the functional dependency between $S_1$ and $S_2$ is delayed we have two cases:
> > > * $S_{1,t} = f(S_{2,t-1},A)$: A can influence S_1 and if it does then CAI would measure it.
> > > * $S_{1,t} = f(S_{2,t-1})$ and $S_{2,t-1}$ is influences by $A$: then our current 1-step CAI implementation would only find influence of $A$ on $S_2$ but not in $S_1$. However, we are thinking about an extension to the multi-step case.

---

> > > > ### Comment · Reviewer_HRby · 2021-09-11
> > > > **Re: response**
> > > >
> > > > Thanks for the clarifications.

---

### Official Review · Reviewer_FgCT · 2021-07-15

**Rating:** 8
**Confidence:** 4

**Summary:**

The authors study how to determine what factors in an environment can be controlled by the agent using a causal setup. One of the contributions is a perspective that causal influence is dependent on situation. They propose to use the conditional mutual information to measure if a future state variable is independent of action conditioned on the current state. Of course, estimating this value is difficult. There are some simplifying assumptions made: that the per state variable transition distribution is Gaussian and various quantities can be predicted by neural networks. After an empirical experiment validating that their Causal Influence Detection approach outperforms other attention-based methods, the authors propose 3 ways in which this detector can be used to improve sample efficiency in RL: 1) better state exploration, 2) causal action exploration and 3) prioritized replay with causal influence.

**Ethical Concerns:**

No.

**Limitations And Societal Impact:**

Yes.

**Main Review:**

originality: Causality and RL are a combination that has received a lot of recent interest, but I found the connection to be well-used in this work to justify the objective. While the objective itself is not particularly novel, the authors derive a way to estimate the quantity in a way that seems to work well empirically.
Some of the three ways in which this quantity is used have certainly been tried before but, again, the combination of the novel measure and the empirical results are compelling.

quality: This paper was a pleasure to read. The experiments are thoughtfully designed to break down the different contributions and show sensitivity to introduced hyper parameters. Baselines are also thoughtfully chosen and explained for most of the experiments. This leads me to some questions and suggestions:

For Figure 4, why was no other exploration method compared to? There are similar intrinsic reward-based exploration methods I would assume to perform similarly, like a count-based method or one based on “surprise,” as measured by epistemic uncertainty via ensembles, or even just simple entropy regularization.

clarity: Same as above. I found the background thoughtfully presented, derivation of the method well explained, and experiments well-designed.

significance: The contributions of this work are substantial. An effective method for computing mutual information quantities in robotics environments will be of great use to RL methods for those applications. The authors have also presented three ways to use that quantity to improve sample efficiency over SOTA in reasonable environments significantly.

**Time Spent Reviewing:**

2

---

> ### Author Response · Authors · 2021-08-09
> **Author response**
>
> We thank you for your review. We are happy that you share our excitement about this work. We would like to point out that Figure 4 is supposed to show a sensitivity analysis of our exploration method to the lambda hyperparameter. In Figure 6, we compare to another exploration method, namely VIME.
> However, as you suggested, we now also added a surprise-based method using epistemic uncertainty from ensembles. We refer to the general answer where we discuss this in more detail.

---

### Official Review · Reviewer_BGJy · 2021-07-16

**Rating:** 7
**Confidence:** 3

**Summary:**

The paper introduces a state-dependent measure of “causal action influence” (CAI) to detect to what extent the agent can affect the next state. The high-level conceptual contribution (as stated by the authors at the beginning of Section 2) is that the measure should in fact be state-dependent. After introducing the measure and a tractable approximation for it, the authors demonstrate various ways in which it can be used to improve the efficiency of learning:
1. Guiding exploration by rewarding the agent for visiting states where it is determined to have more influence
2. Guiding exploration by selecting actions which are determined to have the most influence (among those generated by the policy)
3. A prioritized experience replay sampling scheme which upweights trajectories where the agent visited more states where it has high influence
These approaches are also complementary, and combining the three leads to better results than any alone.

**Limitations And Societal Impact:**

Yes

**Main Review:**

I like the approach of estimating causal influence in the environment. It does seem intuitive that most “real-world” RL tasks would involve causal relationships due to their sequential nature, and uncovering those relationships should lead to more efficient learning.

The CAI estimator presented in the paper has several complementary uses in efficient exploration and learning, so the contribution of the work is clear. The primary downside of the approach is that it requires some domain knowledge, such as a known factorization of the state space and full observability, as well as an idea about which coordinate(s) are likely to contain causal relationships which are useful for exploration. However, in the cases where such information is known, there is a clear benefit to using it.

One suggestion to make the paper even stronger would be to evaluate against another baseline or two for the exploration experiments, e.g. a model-based surprise or random network distillation. The paper mentions a variety of approaches in the related work, but only compares against VIME.

The paper is clear and well-written. I have no significant issues with the writing or organization. The appendix contains useful information such as hyperparameter values and additional plots. I particularly like Figure S4, which plots the CAI over time along with corresponding images of the robot.

**Time Spent Reviewing:**

3

---

> ### Author Response · Authors · 2021-08-09
> **Author response**
>
> We thank you for your review. We agree that some domain knowledge is needed to apply our method. However, we would like to point out that in our RL experiments, we are not using any more domain knowledge than what is given in the standard goal-based RL setting. In particular, goal-based RL methods such as HER assume that the goal-space (e.g. the object position) is given; we use the same goal-space for CAI.
>
> For future work, we also think that the assumption of a factorized state space can be relaxed (e.g. to start from images) if we combine our method with object-centric representation learning such as slot attention [1] or SCALOR [2]. These methods structure the learned state space into objects/entities. We argue that causal relationships between entities are important to explore for many tasks, and as such object-centric representations would naturally give us useful target dimensions to apply CAI.
>
> Thank you for the suggestion of adding more exploration baselines. As per your suggestion, we now ran another exploration method based on model-based surprise. We refer to the general answer where we discuss this in more detail.
>
> [1]: F. Locatello et al., “Object-Centric Learning with Slot Attention,” Advances in Neural Information Processing Systems, vol. 33, 2020.
>
> [2]: J. Jiang, S. Janghorbani, G. de Melo, and S. Ahn, “SCALOR: Generative World Models with Scalable Object Representations,” International Conference on Learning Representations, 2020.

---

### Official Review · Reviewer_VNL6 · 2021-07-18

**Rating:** 6
**Confidence:** 3

**Summary:**

Summary. In reinforcement learning problems, many entities often act independently and only have influence on a small set of other entities. The paper aims to use causality to discover which components of state space are affected by agent’s action and leverage this fact for a more efficient reinforcement learning. To detect cause influence between state component and action, the paper uses mutual information between that component and action (conditioned on current state). Finally, it leverages the detected causal influence to (i) help with exploration by using it as exploration bonus, (ii) take actions with maximum causal influence and (iii) prioritizing experiences with causal influence for training policy. Overall, the paper shows improvement over baseline algorithms in state based robotic tasks.

My main concern is the applicability of the method to image-based settings. I do think latent variable models will cause issues and I am not sure about the extent of the problem. To be fair, authors mention this issue as well in the discussion section. It would be interesting to see how this method performs if state variables are replaced by latent variables (coming from some pretrained generative model; authors can use some expert data for pretraining the generative model for a start to get good features)


**Limitations And Societal Impact:**

It is not addressed in the paper

**Main Review:**

Originality: The paper presents a novel way to incorporate detection of causal influence in state-based RL framework.

Quality: The claims made in the paper are backed up by proper experiments.

Clarity: The paper is clearly written.

Significance: The paper does a good job of showing importance of leveraging causality for sample efficient RL. However, it’s hard to know the long term significance without knowing the applicability of the method to image based setting.


**Time Spent Reviewing:**

1 hr

---

> ### Author Response · Authors · 2021-08-09
> **Author response**
>
> Thank you for your review.
> Your main concern is the broader applicability of our method in an image-based setting. We agree that directly applying our method with a latent variable model could cause problems, namely when the learned latent space is not disentangled/factorized. However, there are also approaches that can obtain such a factorized latent space from images. In particular, we think that our method could work well with object-centric representations obtained by methods like slot attention [1] or SCALOR [2]. Comparing how our method works with different latent variable models is a valuable direction for future work. We would also like to note that access to a factorized state representation is a fairly common assumption in previous work on intrinsic motivation [e.g. 3, 4, 5].
>
> [1]: F. Locatello et al., “Object-Centric Learning with Slot Attention,” Advances in Neural Information Processing Systems, vol. 33, 2020.
>
> [2]: J. Jiang, S. Janghorbani, G. de Melo, and S. Ahn, “SCALOR: Generative World Models with Scalable Object Representations,” International Conference on Learning Representations, 2020.
>
> [3] R. Houthooft et al., “VIME: Variational Information Maximizing Exploration,” Advances in Neural Information Processing Systems, vol. 29, 2016.
>
> [4] S. Blaes, M. Vlastelica Pogančić, J. Zhu, and G. Martius, “Control What You Can: Intrinsically Motivated Task-Planning Agent,” in Advances in Neural Information Processing Systems 32, 2019.
>
> [5] R. Zhao, Y. Gao, P. Abbeel, V. Tresp, and W. Xu, “Mutual Information State Intrinsic Control,” International Conference on Learning Representations, 2020.

---

### Author Response · Authors · 2021-08-09
**General response to all reviewers**

We thank the reviewers for their thoughtful reviews and are happy about their positive evaluation of our paper.

The reviewers pointed out that more baselines for the exploration experiment would be beneficial. We agree and ran an exploration using ensemble disagreement (e.g. [1]) as another baseline, which is a form of model-based surprise. In brief, our CAI exploration is superior to this baseline, also after we improved the baseline method in two ways. More results are below and we will include the corresponding curves and details in the paper.

For the other suggested exploration strategies, we briefly summarize why we see them as not directly applicable in our setting. Count-based strategies are most suitable for discrete state spaces. It would require a discretization of the continuous state space, which is not clear how to do well. Random network distillation is a method designed mainly to work with high-dimensional observation spaces such as images. In essence, it implements a form of model-based surprise similar to ensemble disagreement which we now include. One reviewer also suggested an entropy-regularized strategy. In early experiments, we ran soft-actor critic (SAC) which did not result in a good performance.

Details: The model disagreement [1] baseline did not perform very well (roughly as DDPG+HER with an improved initial period). In this case, the model disagreement on all state variables was used. When modifying it to only take the disagreement on the object coordinates into account (as CAI does), and always using rewards with up-to-date model errors (non-standard), we can obtain better performance that is still slightly worse than exploration with CAI. The following table reports the mean and std. dev. of the success rate (10 seeds) at different times through training as in Figure 4. (In Figure 6 we have accidentally shown the  $\lambda=0.3$ curve which we will correct).

```
| method         \      rollouts            |     4k     |     12k    |     20k    |
| ------------------------------------------|------------|------------|------------|
| model disagreement [1] (best lambda)      | 0.35+-0.03 | 0.55+-0.05 | 0.67+-0.12 |
| modified model disagreement (best lambda) | 0.35+-0.05 | 0.8+-0.12  | 0.82+-0.17 |
| CAI lambda=0.2                            | 0.4+-0.05  | 0.9+-0.05  | 0.97+-0.03 |
```

In particular, model disagreement exhibits notable variance across seeds and seems generally less stable. We believe this is because ensemble disagreement continuously shifts its reward distribution, whereas CAI has a stable target creating the rewards. For both variants, we scanned over 5 different values for the bonus weight hyperparameter.

[1]: D. Pathak, D. Gandhi, and A. Gupta, “Self-Supervised Exploration via Disagreement,” in International Conference on Machine Learning, May 2019

---

### Decision · Program_Chairs · 2021-09-27

**Decision:**

Accept (Poster)

**Comment:**

I would like to congratulate the authors on an excellent investigation. This paper makes very nice progress in the combination of RL and causal inference, and all the reviewers and myself found the work exciting and important.

Please make sure to address the reviewers' feedback in the final version of the paper.